# HOGT: High-Order Graph Transformers

## Abstract

Inspired by the success of transformers on natural language processing (NLP) and computer vision (CV) tasks, graph transformers (GTs) have recently been proposed to boost the performance of graph learning. However, the attention mechanisms used in existing GTs face certain limitations in capturing crucial topological information or scaling to large graphs, due to their quadratic complexity. To address these limitations, in this paper, we propose a high-order information propagation strategy within the transformer architecture to simultaneously learn the local, long-range, and higher-order relationships of the graph. We first propose a flexible sampling method to extract communities from the graph, and create new community nodes and in particular a learnable community sampling method with reinforcement learning. We then propose a three-step message-passing strategy dubbed *HOGT* to capture the local and higher-order information in the communities and propagate long-range dependency information between the community nodes to finally obtain comprehensive node representations. Note that as structural information has been flexibly integrated into our designed community-based message-passing scheme, HOGT discards the positional encoding which was thought to be important for GT. We theoretically demonstrate that GTs with effective substructures can achieve an approximate global attention. HOGT can be viewed as a unified framework, taking many existing graph models as its special cases. We empirically show that HOGT achieves highly competitive results consistently across node and graph classification tasks.

## 1 Introduction

Learning from graph-structured data, such as social networks, biological networks, and brain networks, is critical for many real-world applications. Graph Neural Networks (GNNs) (Kipf & Welling (2017); Veličković et al. (2018); Gasteiger et al. (2019); Hamilton et al. (2017)) are one type of mainstream architecture that adopts a local Message-Passing (MP) scheme where the information is propagated and aggregated between the connected nodes. However, traditional GNNs suffer from the over-smoothing (NT & Maehara (2019)), over-squashing (Topping et al. (2022)), and limited expressiveness (Xu et al. (2019b)) problems because of this neighbourhood-dependent message passing strategy.

The transformer architecture (Vaswani et al. (2017)) has recently attracted great attention for graph learning as its global attention mechanism provides a potential solution to the above problems. In contrast to traditional GNNs, Graph Transformers (GTs) (Kreuzer et al. (2021); Mialon et al. (2021); Ying et al. (2021)) enable information to pass between any two nodes, regardless of the original graph connections. When applying transformers on graphs, the key is to properly incorporate graph structural information. This motivates several studies (Kreuzer et al. (2021); Dwivedi & Bresson (2021); Ying et al. (2021)) to focus on constituting good positional encoding or attention bias to integrate graph structure. However, Müller et al. (2023) showed that current graph transformers still suffer from limited expressivity, and no clear expressivity hierarchy exists for commonly used positional or structural encodings. Moreover, when developing GTs on real graph tasks, especially for node classification, existing models (Chen et al. (2021a); Kreuzer et al. (2021); Park et al. (2022)) suffer from high computational complexity due to dense connections. In conclusion, the current GT models not only fail to fully capture useful topological information (e.g., intrinsic local structure, implicit higher-order correlations) of the graph but also cannot effectively propagate long-range information.

Inspired by the successful use of patches in the vision domain, some recent works (Gao et al. (2022a); Zhao et al. (2023)) have incorporated patch/substructure representations into GTs. While these introduced substructures can benefit graph representation in some cases, existing works (Kuang et al. (2021); Zhu et al. (2023; 2024)) still face challenges in achieving flexible and suitable substructures for different graphs and theoretically demonstration for success of GTs with substructures. Therefore, it is important to develop a new scheme to effectively capture the complex structural relationships in the graph for different graphs and data types, while also providing theoretical support.

In this work, we develop a powerful architecture that can effectively propagate comprehensive information with a flexible sampling method and term it as *HOGT*. To better capture the intricate relationships within a graph, we group graph nodes into multiple communities where all nodes within the same community share similar properties (semantic or information). Notably, we design a learnable community sampling method based on reinforcement learning (RL). When encoding closer graph nodes into the same community, the challenge is how to capture the local high-order information in the community and propagate it globally for effective and comprehensive representation learning. To tackle this challenge, we introduce a new node to represent each community which serves as the bridge to allow the graph node information to propagate and aggregate along these introduced nodes to establish global connections among all nodes. The generated communities can encode more complex structural information as a substitute for positional encoding.

Based on community-structured data, we adopt a three-step message-passing strategy: 1) Graph Node-to-Community Node (*G2C*-**MP**); 2) Community Node-to-Community Node (*C2C*-**ATTN**); and 3) Community Node-to-Graph node (*C2G*-**MP**). In the first step, within each community, the information of each node is propagated and aggregated to its corresponding community node to capture local high-order information. Then, based on the community-level representations of the community nodes, we apply a self-attention mechanism between them to allow each community node to capture long-range information from other communities. Finally, we update the representations of the graph nodes by aggregating information from their respective communities. We can see that the community nodes effectively connect to almost all nodes in the graph.

Our proposed HOGT is a general framework and several other existing graph models can be viewed as special cases. At the level of message-passing strategy: if removing Community Node-to-Community Node (*C2C*-**ATTN**), the framework simplifies to a Message-Passing Architecture. At the level of community generation: if we view the whole graph as a community, our model simplifies to a GT model Wu et al. (2021), which takes a special token to connect with all other nodes to achieve global information, representing the lower bound of HOGT; if we view each node as a community, our model essentially becomes the vanilla transformer, representing the upper bound of HOGT. In comparison to the existing graph models, the advantage of our proposed HOGT in processing various graph information and graph types is shown in Table 1.

Our proposed framework demonstrates its versatility by accommodating various graph types (graph and hypergraph), data types (homophily and heterophily), data scales (same-scale and large-scale), and different graph tasks. We mainly evaluate HOGT on node classification tasks in which GT models have a performance gap, and also extend HOGT for graph classification. We find improvements in accuracy on almost all datasets, especially on heterophilic datasets. In summary, our main contributions are as follows:

- We propose a flexible sampling method followed by a three-step message-passing framework in GTs to capture comprehensive information achieving high expressiveness for graph representation learning.

- We unify message-passing and GTs by constructing communities and introducing new community nodes. We demonstrate that our model can approximate any other message-passing model and theoretically show that the three-step message-passing with newly introduced community node can achieve global attention as general transformers do.

- We conduct extensive experiments on benchmark datasets to demonstrate the effectiveness of the proposed method for node and graph classification. The experimental results also verify the effectiveness of higher-order representations.

Table 1: A summary of the capabilities of different graph models in processing graph information and graph types. GNN is the vanilla graph neural network, HGNN is a hypergraph-based neural network, and GT is the general Graph Transformer.

| Model | Local Information | Global Information | Higher-Order Information | Graph | Hypergraph |
|---|---|---|---|---|---|
| GNN | ✓ | ✗ | ✗ | ✓ | ✗ |
| HGNN | ✓ | ✗ | ✓ | ✓ | ✓ |
| GT | ✓ | ✓ | ✗ | ✓ | ✗ |
| HOGT (ours) | ✓ | ✓ | ✓ | ✓ | ✓ |

## 2 RELATED WORK

**General Graph Transformers.** Recently, the transformer architecture has been successfully applied to the graph domain, showing competitive or even superior performance on many tasks when compared to GNNs. The standard transformer was first extended to graphs (Dwivedi & Bresson (2021)), with four special designs including the position encoding for nodes in a graph. Subsequently, many other GTs (Rong et al. (2020); Zhang et al. (2020); Chen et al. (2021b); Wu et al. (2021); Hussain et al. (2022); Chen et al. (2022a); Nguyen et al. (2022); Kreuzer et al. (2021)) and applications of GTs (Xu et al. (2019a); Zhu et al. (2021; 2022); Cai et al. (2022); Li et al. (2023); Deng et al. (2024); Wu et al. (2024)) have been developed — Rampášek et al. (2022) and Min et al. (2022) provide a more detailed introduction and review of different GTs. However, the above methods are mostly designed for graph-level tasks, as they impose great time and memory constraints due to the self-attention layer. Therefore, several works (Zhao et al. (2021); Choromanski et al. (2022); Guo et al. (2022); Park et al. (2022); Wu et al. (2023); Liu et al. (2023)) have been proposed to make graph transformers more scalable and efficient, but they still suffer from various challenges such as missing long-range and higher-order information or noise aggregation.

**Graph Transformer Utilizing Substructures.** Due to the exponentially increasing scale of graph data, researchers have attempted to utilize substructures to scale up graph representation learning through methods such as subgraph learning (Kim & Oh), and graph condensation (Wang et al. (2024); Zheng et al. (2024); Zhou et al. (2023); Jin et al. (2021); Huang et al. (2021); Fey et al. (2020)). In terms of GTs, substructures (Zhang et al. (2022b)) (such as hierarchical structure, clusters, communities, and subgraphs) have been utilized for both graph and node classification. For graph classification tasks, some methods (Gao et al. (2022a); Zhao et al. (2023)) segment the graph into patches or subgraphs and use the substructural representations to learn topological high-level information. For node classification, researchers have explored extracting substructures and utilizing these substructural representations to reduce the quadratic complexity of global self-attention, while effectively capturing the global information of the graph. Specifically, Kuang et al. (2021) and Zhu et al. (2023) employed coarsening techniques to obtain a coarser graph with fewer nodes to capture long-range information. With the obtained clusters, Xing et al. (2024) introduced inter-cluster and intra-cluster Transformers to extract local information and long-range dependent information from distant nodes. Zhu et al. (2024) and Jiang et al. (2023) selected several topologically important nodes as anchors, allowing information to propagate over a large receptive field, where the anchor nodes can be viewed as a substructure of the original graph. Furthermore, Fu et al. (2024) transferred global and long-range information by establishing multiple virtual connections using personalized PageRank.

While these introduced substructures can benefit graph representation in some cases, there are limitations of existing works for achieving flexible and suitable substructures for different graphs. Specifically, the anchor nodes in Zhu et al. (2024); Jiang et al. (2023) are derived from the original graph nodes and therefore cannot introduce additional information, such as higher-order information. Similarly, the supernodes in the coarse graphs in Kuang et al. (2021); Zhu et al. (2023) which are used to propagate high-level information are also constrained by the original graph structure. Moreover, it is not trivial to provide theoretical support for GTs with substructures in capturing global attention, while ensuring adaptability across various graph datasets. In this work, we demonstrate that the proposed HOGT offers a general and theoretically grounded framework. It shows significant advantages in capturing comprehensive information through the flexible community sampling method and demonstrates its versatility by its effectiveness across diverse graph datasets.

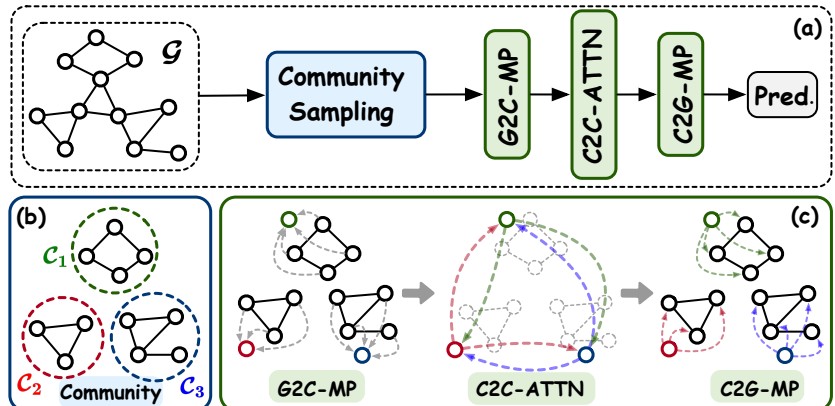

Figure 1: (a) The overall HOGT framework. (b) HOGT first adopts a community sampling method to obtain multiple communities. (c) Then, it propagates and aggregates information following a three-step operation: 1) **G2C**-**MP** which aggregates the high-order information of a community into the community node; 2) **C2C**-**MP** which propagates community-level information in a self-attention mechanism; 3) **C2G**-**MP** which gathers the updated community-level information for node representation.

More related works about Higher-Order Representation Learning and Virtual Node in Message-Passing can be found in Appendix A.1.

## 3 HIGH-ORDER GRAPH TRANSFORMER

**Overview.** As illustrated in Figure 1, our proposed HOGT framework is designed to effectively aggregate and propagate all levels of information for comprehensive graph representation learning. By dividing the whole graph into several communities and introducing a representative node for each community, we achieve the local higher-order representation of each community and adopt community-level attention to effectively propagate the long-range dependent information. In the following, first we introduce the notations and provide a background on transformer architecture, and then we describe in detail each component of the architecture. The complexity analysis of HOGT can be found in Appendix A.2.

**Notation.** Given an graph $\mathcal{G} = (\mathcal{V}, \mathcal{E})$ with node set $\mathcal{V}$ and edge set $\mathcal{E}$. Suppose there are $N$ nodes in $\mathcal{V}$, the set of edges $\mathcal{E} \subseteq \mathcal{V} \times \mathcal{V}$ defines the connections among the $N$ nodes, $(v_i, v_j) \in \mathcal{E}$ denotes the edge between node $v_i$ and node $v_j$. The graph topology is presented by the adjacency matrix $\mathbf{A}$, where $\mathbf{A}_{ij} = 1$ if there exists an edge $(v_i, v_j)$, $\mathbf{A}_{ij} = 0$ otherwise. We denote $\mathbf{X} \in \mathbb{R}^{N \times d}$ the node features, where each node $i$ has $x \in \mathbb{R}^d$. Let $y_i$ denote the label of node $i$, in this work, we focus on the node classification task which aims to predict the labels of the unknown nodes in the graph.

**Transformer Architecture.** The transformer architecture consists of a composition of transformer layers. Each transformer layer has a self-attention module and a position-wise feed-forward network (FFN). The self-attention mechanism calculates attention scores by taking the inner product of query vectors ($\mathbf{Q}$) and key vectors ($\mathbf{K}$). It then uses these scores to aggregate value vectors ($\mathbf{V}$) in a weighted manner, resulting in contextualized representations, that is,

$$\mathbf{Q} = \mathbf{H}\mathbf{W}^Q, \mathbf{K} = \mathbf{H}\mathbf{W}^K, \mathbf{V} = \mathbf{H}\mathbf{W}^V; \tag{1}$$

$$\mathbf{A} = \frac{\mathbf{Q}\mathbf{K}^\top}{\sqrt{d'}}, \quad \mathrm{Attn}(\mathbf{H}) = \mathrm{softmax}(\mathbf{A})\mathbf{V} \tag{2}$$

where $\mathbf{W}^Q \in \mathbb{R}^{d \times d'}, \mathbf{W}^K \in \mathbb{R}^{d \times d'}$, and $\mathbf{W}^V \in \mathbb{R}^{d \times d'}$ are projection matrices, $\mathbf{H} = \left[\boldsymbol{h}_1^\top, \ldots, \boldsymbol{h}_n^\top\right]^\top \in \mathbb{R}^{n \times d}$ denotes the input matrix of node embeddings, and $d'$ is the output hidden dimension. Generally, it is the global attention mechanism that allows everything to connect to everything (e.g., every node to every node for graph). Instead of performing a single attention function, it is standard to adopt multi-head attention.

### 3.1 COMMUNITY SAMPLING

Effectively utilizing the structural information of the graph is the key challenge for graph representation learning. We note that data correlations in practice can be complex and are often beyond pairwise, for example, a community of friends shares their common interest in basketball in a social network. To encode these higher-order correlations, we consider extracting meaningful communities from the whole graph. Here, a community is introduced to collect multiple vertices sharing similar properties (semantic or information), similar to how a hyperedge connects multiple objects.

We design the community sampling method tailored to different graph types: 1). For hypergraphs, we intuitively view each hyperedge as a community; 2). For regular graphs, we explore a learnable sampling method that employs reinforcement learning to determine the optimal communities.

**Learnable sampling.** For graph $\mathcal{G} = \{\mathbf{A}, \mathbf{X}\}$, we learn a GNN-based encoder and obtain the hidden representation of $N$ nodes: $\mathbf{H} = \left[\boldsymbol{h}_1^\top, \ldots, \boldsymbol{h}_N^\top\right]^\top \in \mathbb{R}^{N \times d}$. Then, we employ a trainable projection vector $\mathbf{p}$ to project all node features to 1D. Given node $v_i$ with feature $\boldsymbol{h}_i$, its scale projection on $\mathbf{p}$ is $y_i = \boldsymbol{h}_i \mathbf{p}/\|\mathbf{p}\|$.. Here, $y_i$ measures how much information of node $v_i$ can be retained when projected to the direction of $\mathbf{p}$. After that, we adopt top-$k$ sampling to select $kN$ nodes, where $k \in (0, 1]$. For each selected node $i$, we generate a community $\tilde{\mathcal{V}}_i$ with its neighbors. To find the optimal $k$ in top-$k$ sampling, we present a reinforcement learning (RL) algorithm to update the sampling ratio $k$ adaptively. We model the updating process of $k$ as a finite horizon Markov Decision Process (MDP) and adopt Q-learning Watkins & Dayan (1992); Sun et al. (2021) to learn the MDP. In the experiments, we also apply two general sampling approaches: random walk Zeng et al. (2019) and spectral clustering Chiang et al. (2019). More details of these three sampling methods can be found in Appendix A.3.

### 3.2 MODEL DESIGN

Operating on communities, HOGT leverages the following three steps to obtain local, high-order, and long-range information. While achieving expressive representation with reduced computational cost, the proposed community-based method can be viewed as a new structural encoding strategy.

**(1) Graph Node-to-Community Node (*G2C*-MP).** To capture the higher-order information in the community, we introduce a representative community node (CN) for each community, and connect it with other nodes in the community. The use of an additional node (virtual node) that connects to all input graph nodes, has been observed to improve GNNs (Gilmer et al. (2017); Hu et al. (2021a); Wu et al. (2021)) and has been justified theoretically (Cai et al. (2023)). Instead of aggregating the whole information in the graph (as READOUT) (Wu et al. (2021)), we introduce the additional node for each community to capture the higher-order structural information of a graph and support the global information propagation as bridges.

Assume there are $m$ communities $\left\{\tilde{\mathcal{V}}_1, \ldots, \tilde{\mathcal{V}}_m\right\}$, we then have $m$ community nodes $\overline{V} = \{\overline{v}_1, \ldots, \overline{v}_m\}$. We initialize the community node feature $\overline{x}_i$ with a $d$-dimensional random vector. Note that the number of community nodes is significantly smaller than the number of graph nodes. For each community $\tilde{\mathcal{V}}_i$, the community representation can be obtained by the community node acting as the query $\overline{\boldsymbol{q}}_i$ with $\overline{\boldsymbol{q}}_i = \overline{x}_i \mathbf{W}^Q$:

$$\boldsymbol{h}_i^c = \text{softmax}\left(\alpha \overline{\boldsymbol{q}}_i \mathbf{K}_{\tilde{\mathcal{V}}_i}^\top\right) \mathbf{V}_{\tilde{\mathcal{V}}_i}, \tag{3}$$

where $\alpha$ is a constant scalar ($\alpha = \frac{1}{\sqrt{d'}}$), $\mathbf{K}_{\tilde{\mathcal{V}}_i}$ and $\mathbf{V}_{\tilde{\mathcal{V}}_i}$ are the key and value matrices of $\overline{v}_i$'s community. The community node aggregates the community-level information.

**(2) Community Node-to-Community Node (*C2C*-ATTN).** To maintain the benefit of global attention in the transformer architecture, we enable information propagation between any two communities. Viewing each community node as a token, we adopt self-attention to refine the community-level representations:

$$\text{Attn}(\mathbf{H}^c) = \text{softmax}(\frac{\mathbf{Q}^c \mathbf{K}^{c\top}}{\sqrt{d'}})\mathbf{V}^c, \tag{4}$$

where $\mathbf{Q}^c = \mathbf{H}^c \mathbf{W}^Q$, $\mathbf{K}^c = \mathbf{H}^c \mathbf{W}^K$, $\mathbf{V}^c = \mathbf{H}^c \mathbf{W}^V$, with $\mathbf{H}^c = \left[ \boldsymbol{h}_1^{c\top}, \ldots, \boldsymbol{h}_m^{c\ \top} \right]^\top \in \mathbb{R}^{m \times d'}$. By propagating information between communities, we obtain the updated community representation $\mathbf{H}^{c'}$. The information passing from community to community helps to: 1) enhance the relationship of communities, and 2) capture the long-range dependency at the community level.

**(3) Community Node-to-Graph Node (*C2G*-MP).** To finally obtain the representation of each node, we aggregate the community representations to update node features. We define the query vector of graph node $v_i$ as $\boldsymbol{q}_i$, while the key and value matrices from introduced community nodes are $\mathbf{K}^{c'} \in \mathbb{R}^{m \times d'}$ and $\mathbf{V}^{c'} \in \mathbb{R}^{m \times d'}$, respectively. For graph node $v_i$, its representation can be enhanced with community-level representations as:

$$\boldsymbol{h}_i = \mathrm{softmax}\left( \alpha \boldsymbol{q}_i \mathbf{K}_{V(i)}^{c'}{}^\top \right) \mathbf{V}_{V(i)}^{c'}, \tag{5}$$

where $\mathbf{K}_{V(i)}^{c'}$ and $\mathbf{V}_{V(i)}^{c'}$ are the key and value matrices of $v_i$'s communities.

Considering the importance of neighbors, it is also necessary to maintain local message-passing (Zhao et al. (2021)) for the local-dependency graph data. Thus, the representation of graph node $v_i$ can be updated as follows:

$$\boldsymbol{h}_i = \mathrm{softmax}\left( \alpha \boldsymbol{q}_i \mathbf{K}_{V(i)}^\top \right) \mathbf{V}_{V(i)}, \tag{6}$$

where $\mathbf{K}_{V(i)} = \begin{bmatrix} \mathbf{K}_{V(i)}^{c'} \\ \mathbf{K}_{\mathcal{N}(i)} \end{bmatrix}$ is the combination of $\mathbf{K}_{V(i)}^{c'}$ and $\mathbf{K}_{\mathcal{N}(i)}$, and $\mathbf{V}_{V(i)}$ is the combination of $\mathbf{V}_{V(i)}^{c'}$ and $\mathbf{V}_{\mathcal{N}(i)}$, where $\mathbf{K}_{\mathcal{N}(i)}, \mathbf{V}_{\mathcal{N}(i)}$ are the key and value matrices of neighboring nodes of $v_i$, respectively.

**Implementation Details of HOGT**   We have presented the individual attention mechanism in line with general transformers. HOGT adopts multi-head attention (MHA) followed by feed-forward blocks (FFN) and layer normalization (LN$(\cdot)$) as:

$$\boldsymbol{h}'^{(l)} = \mathrm{LN}\left( \mathrm{MHA}\left( \boldsymbol{h}^{(l-1)} \right) \right) + \boldsymbol{h}^{(l-1)}; \quad \boldsymbol{h}^{(l)} = \mathrm{LN}\left( \mathrm{FNN}\left( \boldsymbol{h}'^{(l)} \right) \right) + \boldsymbol{h}'^{(l)}. \tag{7}$$

The positional encoding is an important component in transformers, and in the graph domain, researchers have integrated the positional information into GTs by random walk positional encoding Dwivedi & Bresson (2021), or Laplacian positional encoding Dwivedi et al. (2021). In HOGT, the proposed community-based method can be viewed as a new structural encoding strategy.

## 4 THEORETICAL ANALYSIS

Here, we analyze several properties of HOGT including 1) the lower bound of HOGT, 2) the upper bound of HOGT, and 3) a general case of HOGT. We show that HOGT is a powerful model that can approximate the GT model and achieve global attention, i.e., unifying MP and GT with the community and newly introduced community nodes. We also analyze the role of community nodes in capturing the high-order representation versus the function of hyperedges in hypergraph convolutional networks in Appendix A.4.

**Viewing the Whole Graph as a Community.**   In this case, GT can be simplified by Message-Passing Neural Networks (MPNN) with an additional node that connects to all graph nodes. This forms the lower bound of HOGT (number of communities $m = 1$). It has been demonstrated by Cai et al. (2023) that MPNN with a virtual node can approximate a self-attention layer arbitrarily well.

**Viewing Each Node as a Community.**   In this case, HOGT is the standard transformer. Specifically, the three-step MP in HOGT is reduced to one step: Community Node-to-Community Node. Since a node is a community, HOGT is equivalent to propagating information between any two nodes.

**Multiple Communities With Multiple Nodes.** In the general case, there are multiple communities with each containing multiple nodes. In this case, we demonstrate the power of HOGT by showing that the information passing from graph nodes to community nodes back to graph nodes can approximate global attention arbitrarily well.

**Definition 4.1.** A full self-attention layer is defined as:

$$
\boldsymbol{x}_i^{(l+1)} = \sum_{j=1}^{n} \frac{\phi\left(\boldsymbol{q}_i\right)^T \phi\left(\boldsymbol{k}_j\right)}{\sum_{k=1}^{n} \phi\left(\boldsymbol{q}_i\right)^T \phi\left(\boldsymbol{k}_k\right)} \cdot \boldsymbol{v}_j = \frac{\left(\phi\left(\boldsymbol{q}_i\right)^T \sum_{j=1}^{n} \phi\left(\boldsymbol{k}_j\right) \otimes \boldsymbol{v}_j\right)^T}{\phi\left(\boldsymbol{q}_i\right)^T \sum_{k=1}^{n} \phi\left(\boldsymbol{k}_k\right)}, \tag{8}
$$

where $\phi(\cdot)$ is a low-dimensional feature map with random transformation, $\boldsymbol{q}_i$, $\boldsymbol{k}_i$, $\boldsymbol{v}_i$ are the query, key, and value vector, respectively.

**Proposition 4.1.** *The $\sum_{k=1}^{n} \phi\left(\boldsymbol{k}_k\right)$ and $\sum_{j=1}^{n} \phi\left(\boldsymbol{k}_j\right) \otimes \boldsymbol{v}_j$ can be approximated by the virtual node, and shared for all graph nodes, using only $\mathcal{O}(1)$ layers of MPNNs.*

Proposition 4.1 asserts that Message-Passing Neural Networks with community nodes (MPNN+CN) can function as the self-attention layer. Based on Proposition 4.1, we derive the following theorem for our three-step message-passing framework.

**Theorem 4.1.** *The combination of Message-Passing and self-attention: Message-Passing with an introduced new node followed by a self-attention aggregation followed by another Message-Passing can approximate self-attention arbitrarily well.*

We briefly show how the approximation error can be bounded in Proposition 4.1 and provide the proof of Theorem 4.1 in Appendix A.5.

## 5 EXPERIMENTS

In this section, we evaluate the effectiveness of HOGT in node classification tasks, a scenario where GTs have yet to demonstrate state-of-the-art performance. We compare HOGT with standard GCN-based models (graph and hypergraph-based), heterophilic-graph based models, and GT-based models. We also apply HOGT for graph classification task on datasets from TU database (Morris et al. (2020)) and link prediction task on TEG-DB datasets (Li et al. (2024)) to further demonstrate its superiority in Appendix A.9. Then, we evaluate the components of HOGT, including community sampling; structural encoding; the necessity of message-passing between the communities, the local connections in the graph. The detailed experiment settings can be found in Appendix A.6. The analysis of community node initialization is included in the Appendix A.7. Additional analyses and results for the HOGT model, including hyperparameter sensitivity and robustness evaluations, can be found in Appendices A.10 and A.11, respectively.

**Datasets** We experiment on a range of graph benchmarks: (1) homophilic graph datasets (Cora, Citeseer, Pubmed, and ogbn-arxiv) (Pei et al. (2020); Hu et al. (2020b)), (2) heterophilic graph datasets (Cornell, Texas, Wisconsin, Actor, roman-empire, and amazon-ratings) (Zhu et al. (2020); Platonov et al. (2023)), and (3) hypergraph datasets (Co-authorship Cora, DBLP, and News20) (Zhou et al. (2006); Yadati et al. (2019); Chien et al. (2021a)), involving diverse domains and sizes (roman-empire, amazon-ratings, Co-authorship DBLP, and ogbn-arxiv are large-scale datasets). The details of these datasets are provided in Appendix A.8.

### 5.1 MAIN RESULTS

**Performance on Homophilic Graphs.** The homophilic datasets are graphs with high **Homo.** (indicating the proportion of edges connecting nodes with the same label (Zhu et al. (2020))). The prediction accuracies for node classification tasks are reported in Table 2. It can be observed that our proposed HOGT method achieves the state-of-the-art or a competitive performance on most of the datasets, regardless of the sampling method.

Compared with GCN-based methods, HOGT performs better on graphs with more nodes (*e.g.*, Pubmed), specifically, HOGT improves upon popular GNN methods-APPNP by a margin of 3% on Pubmed. This is likely because, based on local message-passing, GCN methods only capture local

Table 2: Node classification results on different datasets (mean accuracy (%) and standard deviation over 10 different runs). **Red**: the best performance per dataset. **Blue**: the second best performance per dataset. OOM denotes out-of-memory.

| | Cora | Citeseer | Pubmed | ogbn-arxiv |
|---|---|---|---|---|
| *GCN-based methods* | | | | |
| GCN Kipf & Welling (2017) | $86.92_{\pm1.33}$ | $76.13_{\pm1.51}$ | $87.01_{\pm0.62}$ | $70.40_{\pm0.10}$ |
| APPNP Gasteiger et al. (2019) | $87.75_{\pm1.30}$ | $\textbf{76.53}_{\pm1.61}$ | $86.52_{\pm0.61}$ | $70.20_{\pm0.16}$ |
| GCNII Chen et al. (2020) | $86.08_{\pm2.18}$ | $74.75_{\pm1.76}$ | $85.98_{\pm0.61}$ | $69.78_{\pm0.16}$ |
| GAT Veličković et al. (2018) | $87.34_{\pm1.14}$ | $75.75_{\pm1.86}$ | $85.37_{\pm0.56}$ | $67.56_{\pm0.12}$ |
| GATv2 Brody et al. (2022) | $87.25_{\pm0.89}$ | $75.72_{\pm1.30}$ | $85.75_{\pm0.55}$ | $68.84_{\pm0.13}$ |
| HGNN Feng et al. (2019) | $86.88_{\pm1.22}$ | $75.87_{\pm1.47}$ | $84.71_{\pm0.56}$ | OOM |
| HGNN+ Gao et al. (2022b) | $83.22_{\pm0.91}$ | $74.71_{\pm1.64}$ | $83.77_{\pm0.65}$ | – |
| *Graph Transformer-based methods* | | | | |
| SAN Kreuzer et al. (2021) | $81.91_{\pm3.42}$ | $69.63_{\pm3.76}$ | $81.79_{\pm0.98}$ | $69.17_{\pm0.15}$ |
| Graphormer Ying et al. (2021) | $67.71_{\pm0.78}$ | $73.30_{\pm1.21}$ | OOM | OOM |
| LiteGT Chen et al. (2021a) | $80.62_{\pm2.69}$ | $69.09_{\pm2.03}$ | $85.45_{\pm0.69}$ | OOM |
| UniMP Shi et al. (2020) | $84.18_{\pm1.39}$ | $75.00_{\pm1.59}$ | $88.56_{\pm0.32}$ | $\textbf{73.19}_{\pm0.18}$ |
| ANS-GT Zhang et al. (2022a) | $86.71_{\pm1.45}$ | $74.57_{\pm1.51}$ | $\textbf{89.76}_{\pm0.46}$ | – |
| NodeFormer Wu et al. (2022) | $86.00_{\pm1.59}$ | $76.70_{\pm1.70}$ | $88.76_{\pm0.50}$ | – |
| Gapformer Liu et al. (2023) | $87.37_{\pm0.76}$ | $76.21_{\pm1.47}$ | $88.98_{\pm0.46}$ | $\textbf{71.90}_{\pm0.19}$ |
| HOGT (randomwalk) | $\textbf{88.11}_{\pm1.05}$ | $\textbf{76.74}_{\pm1.47}$ | $89.20_{\pm1.34}$ | $71.38_{\pm0.14}$ |
| HOGT (clustering) | $88.09_{\pm1.34}$ | $76.35_{\pm1.47}$ | $88.96_{\pm0.49}$ | $71.10_{\pm0.72}$ |
| HOGT (learnable) | $\textbf{88.53}_{\pm1.26}$ | $\textbf{77.59}_{\pm0.94}$ | $\textbf{89.52}_{\pm0.55}$ | $\textbf{72.02}_{\pm0.25}$ |

structural information. By contrast, HOGT enables the learning of more informative representations, including community- and global-level information, which represents a significant advantage.

Compared to GT-based methods, there is an obvious advantage for HOGT on small-scale datasets (*e.g.*, Cora and Citeseer) with higher **Homo.**, i.e., where the local-neighborhood information is more important. Thus, the vanilla global attention on the whole graph adopted in existing GTs (such as Graphormer) leads to massive unrelated information aggregation. Gapformer, a special case of HOGT with one community, also achieves good performance.

In terms of efficiency, HOGT can be easily applied to large-scale graphs, ogbn-arxiv, while some other GT methods cannot due to their poor scalability. Particularly, Graphformer and LiteGT encountered out-of-memory errors, even on small graphs. This highlights the need for a GT that can scale effectively to large-scale graphs.

**Performance on Heterophilic Graphs.** These heterophilic datasets are of low **Homo.**, thus can be viewed as long-range dependency datasets. From the results in Table 3, we can observe that specially designed heterophily-based methods can generally achieve improved performance, but not on large-scale datasets (roman-empire, amazon-ratings). Except for Gapformer, most GT-based models demonstrate a poor performance, which implies that GTs fail to propagate and aggregate useful information. By contrast, our HOGT method can be easily extended to heterophilic graph datasets. Specifically, for 4 small-scale datasets, HOGT improves upon the popular heterophily-based GNN method GPRGNN by margins of 2.7%, 4.6%, 2.8% (absolute differences) on Cornell, Wisconsin, and Actor. Compared to Gapformer, HOGT achieves performance gains of 2.1%, 3.1%, 3.7% on Cornell, Texas, and Wisconsin, respectively. On the 2 large-scale heterophilic datasets (roman-empire and amazon-ratings), HOGT is significantly better than previous models. We further evaluate the effectiveness of HGT by t-test in Appendix A.9 and find that the improvements of HGT over baselines are all statistically significant (p-value$\ll$0.05).

Table 3: Node classification results on heterophilic datasets (mean accuracy (%) and standard deviation over 10 different runs). **Red**: the best performance per dataset. **Blue**: the second best performance per dataset.

| | Cornell | Texas | Wisconsin | Actor | roman-empire | amazon-ratings |
|---|---|---|---|---|---|---|
| *GCN-based methods* | | | | | | |
| GCN Kipf & Welling (2017) | $45.67_{\pm7.96}$ | $60.81_{\pm8.03}$ | $52.55_{\pm4.27}$ | $28.73_{\pm1.17}$ | $73.69_{\pm0.74}$ | $48.70_{\pm0.63}$ |
| APPNP Gasteiger et al. (2019) | $41.35_{\pm7.15}$ | $61.62_{\pm5.37}$ | $55.29_{\pm3.90}$ | $29.42_{\pm0.81}$ | $72.73_{\pm0.44}$ | $45.62_{\pm0.52}$ |
| GAT Veličković et al. (2018) | $47.02_{\pm7.66}$ | $62.16_{\pm4.52}$ | $57.45_{\pm3.51}$ | $28.33_{\pm1.13}$ | $80.87_{\pm0.30}$ | $49.09_{\pm0.63}$ |
| GATv2 Brody et al. (2022) | $50.27_{\pm8.97}$ | $60.54_{\pm4.55}$ | $52.74_{\pm3.96}$ | $28.79_{\pm1.47}$ | $80.99_{\pm0.98}$ | $44.00_{\pm0.67}$ |
| *Heterophily-based methods* | | | | | | |
| MLP LeCun et al. (2015) | $71.62_{\pm5.57}$ | $77.83_{\pm5.24}$ | $82.15_{\pm6.93}$ | $33.26_{\pm0.91}$ | $64.45_{\pm0.61}$ | $42.44_{\pm0.70}$ |
| MixHop Abu-El-Haija et al. (2019) | $76.48_{\pm2.97}$ | $83.24_{\pm4.48}$ | $85.48_{\pm3.06}$ | $34.92_{\pm0.91}$ | $82.90_{\pm0.57}$ | $51.35_{\pm0.38}$ |
| H2GCN Zhu et al. (2020) | $75.40_{\pm4.09}$ | $79.73_{\pm3.25}$ | $77.57_{\pm4.11}$ | $36.18_{\pm0.45}$ | $60.11_{\pm0.52}$ | $36.47_{\pm0.23}$ |
| FAGCN Bo et al. (2021) | $67.56_{\pm5.26}$ | $75.67_{\pm4.68}$ | $75.29_{\pm3.06}$ | $32.13_{\pm1.33}$ | $65.22_{\pm0.56}$ | $44.12_{\pm0.30}$ |
| GPRGNN Chien et al. (2021b) | $76.76_{\pm2.16}$ | $81.08_{\pm4.35}$ | $82.66_{\pm5.62}$ | $35.30_{\pm0.80}$ | $64.85_{\pm0.27}$ | $44.88_{\pm0.34}$ |
| *Graph Transformer-based methods* | | | | | | |
| SAN Kreuzer et al. (2021) | $50.85_{\pm8.54}$ | $60.17_{\pm6.66}$ | $51.37_{\pm3.08}$ | $27.12_{\pm2.59}$ | OOM | OOM |
| UniMP Shi et al. (2020) | $66.48_{\pm12.5}$ | $73.51_{\pm8.44}$ | $79.60_{\pm5.41}$ | $35.15_{\pm0.84}$ | - | - |
| NAGphormer Chen et al. (2022b) | $56.22_{\pm8.08}$ | $63.51_{\pm6.53}$ | $62.55_{\pm6.22}$ | $34.33_{\pm0.94}$ | $76.12_{\pm0.22}$ | $49.44_{\pm0.54}$ |
| Gapformer Liu et al. (2023) | $77.57_{\pm3.43}$ | $80.27_{\pm4.01}$ | $83.53_{\pm3.42}$ | $36.90_{\pm0.82}$ | $87.65_{\pm0.47}$ | $46.38_{\pm0.58}$ |
| HOGT (randomwalk) | $79.46_{\pm2.16}$ | $83.44_{\pm1.87}$ | $87.25_{\pm2.67}$ | $38.11_{\pm0.87}$ | $88.74_{\pm0.52}$ | $53.94_{\pm0.43}$ |
| HOGT (clustering) | $78.65_{\pm2.82}$ | $82.63_{\pm4.97}$ | $86.47_{\pm2.97}$ | $37.44_{\pm0.68}$ | $88.47_{\pm0.53}$ | $53.59_{\pm0.59}$ |
| HOGT (learnable) | $79.73_{\pm3.25}$ | $81.62_{\pm4.49}$ | $85.10_{\pm2.00}$ | $38.62_{\pm1.02}$ | $88.94_{\pm0.52}$ | $54.32_{\pm0.44}$ |

Table 4: Analysis of positional encoding on different datasets (mean accuracy (%) and standard deviation over 10 different runs).

| Community Sampling | Model | Cora | Citeseer | Cornell | Texas | Wisconsin |
|---|---|---|---|---|---|---|
| Spectral Clustering | HOGT(lpe) | 87.79±1.33 | 75.87±1.75 | 71.35±4.05 | 77.30±7.37 | 81.96±3.26 |
| | HOGT(rwpe) | 87.52±1.53 | 75.65±1.78 | 73.78±3.83 | 78.38±4.01 | 84.71±2.11 |
| | HOGT(w/o pe) | 88.09±1.34 | 76.35±1.47 | 78.65±2.82 | 82.63±4.97 | 86.47±2.97 |

More experimental results in Appendix 10 show that HOGT achieves better performance than popular hypergraph methods HGNN Feng et al. (2019) and HGNN+ Gao et al. (2022b) across all hypergraph datasets. Compared to traditional HGCN methods, HOGT can propagate higher-order information more flexibly based on attention architecture.

## 5.2 EFFICIENCY AND SCALABILITY

We evaluate the scalability of the proposed HOGT on two other large-scale datasets, ogbn-proteins and ogbn-products Hu et al. (2020b). As shown in Table 5, HOGT outperforms all the baselines on these large graphs. Table 6 reports the training time per epoch, inference time, and GPU memory costs for Cora and ogbn-proteins. Since it is common practice to use a fixed number of training epochs for model training on these datasets, we report the training time per epoch to compare training efficiency. We observe that HOGT is orders of magnitude faster than popular GT models, including Graphormer Ying et al. (2021), LiteGT Chen et al. (2021a), and Polynormer Deng et al. (2024). Compared to GAT, APPNP, and SG-Former Wu et al. (2024), HOGT strikes a balance between performance and efficiency.

Table 5: Node classification results on large-scale datasets (mean accuracy (%) and standard deviation over 3 different runs).

| Model | ogbn-proteins | ogbn-products |
|---|---|---|
| Graphormer | OOM | OOM |
| SAN | OOM | OOM |
| ANS-GT | $74.67 \pm 0.65$ | $80.64 \pm 0.29$ |
| HSGT | $78.13 \pm 0.25$ | $81.15 \pm 0.13$ |
| SGFormer | $79.53 \pm 0.38$ | $81.61 \pm 0.26$ |
| Polynormer | $78.20 \pm 0.44$ | $82.97 \pm 0.28$ |
| HOGT | $80.39 \pm 0.64$ | $83.48 \pm 0.32$ |

## 5.3 Ablation Studies

Here, we conduct a set of ablation studies to test different configurations of HOGT. The effect of self-attention between communities and local information can be found in Appendix A.9.

**Evaluation with Community Sampling.**  Here, we compare the performance of HOGT with three different community sampling methods, i.e., learnable, random walk and spectral clustering. The results have been included in Tables 2 and 3. It shows that HOGT with proposed learnable sampling slightly outperforms random walk while random walk sampling slightly outperforms spectral clustering. Intuitively, 1) random walk sampling constrains the nodes in a community with $k$-hop walk length, while spectral clustering separates the graph from a global view. Thus, random walk sampling captures more local structural information than spectral clustering method; 2) Spectral clustering method is more sensitive to data types (homogeneous or heterogeneous) than random walk, as it focuses on global connections. A larger difference between HOGT (randomwalk) and HOGT (clustering) is observed on heterophilic datasets compared to homophilic datasets. It is important to note that the results of HOGT (clustering) on large-scale heterophilic datasets (roman-empire and amazon-ratings) are reported with a single community. Increasing the number of communities will result in a significant performance decrease. While our proposed learnable method can actively select optimal communities, HOGT (learnable) can achieve improved performance. We further analyze the effect of the number of communities on two unlearnable sampling methods in Appendix A.9.

Table 6: Efficiency comparison of HOGT and graph Transformer competitors w.r.t. training time per epoch, inference time and GPU memory (GB) cost on a A100. The missing results are caused by out-of-memory.

| Method | Cora | | | ogbn-proteins | | |
|---|---|---|---|---|---|---|
| | Tr (ms) | Inf (ms) | Mem (MB) | Tr (s) | Inf (s) | Mem (MB) |
| GAT | 3.18 | 1.68 | 166.35 | - | - | - |
| APPNP | 3.32 | 1.49 | 35.57 | - | - | - |
| Graphormer | 90.58 | 71.26 | 359.25 | - | - | - |
| LiteGT | 15.57 | 5.77 | 227.69 | - | - | - |
| polynormer | 218.23 | 5.13 | 264.06 | 1.60 | 0.127 | 6429.06 |
| SGFormer | 3.66 | 1.42 | 50.87 | 1.26 | 0.098 | 228.19 |
| HOGT | 7.40 | 2.69 | 109.22 | 1.12 | 0.087 | 1284.32 |

**Effect of Position Encoding.**  Based on Spectral Clustering, we test the role of positional encoding for the proposed HOGT. We compare two popular positional encoding methods including Laplacian-based (lpe) and random walk positional encoding (rwpe) to HOGT without any positional encoding (w/o pe). It can be seen from Table 4 that the gap in performance is minor with or without positional encoding on homophilic datasets (Cora and Citeseer). While without positional encoding, HOGT achieves obvious better performance on heterophilic datasets, such as, Cornell, Texas, and Wisconsin. The positional encoding methods (such as lpe) usually encode the original graph connections, thus, integrating positional encoding will lead to a negative effect for these heterophilic datasets which contain massive noisy information in graph structure. A detailed analysis of the failure of positional encoding can be found in Appendix A.9. Compared to popular positional encoding methods, community sampling in HOGT are able to integrate structural information in a more flexible and effective way.

## 6 Conclusion

In this paper, we introduced a higher-order message-passing strategy within the Transformer architecture to learn long-range, higher-order relationships for graph representation. Initially, we extract communities from the entire graph and introduce a new node for each community. Subsequently, leveraging community-structured data, we adopt a three-step message-passing scheme to aggregate information from the graph node to the community node, propagate information between community nodes and send the community-level information back to the graph nodes. The introduced nodes act like hyperedges in a hypergraph to effectively propagate information to other graph nodes. We theoretically demonstrate the powerful expressiveness of HOGT and empirically show the effectiveness of HOGT across diverse datasets on node classification. While HOGT is designed to capture comprehensive information across various types of graphs, achieving an optimal balance between different (local, global, and higher-order) information in complex graph structures remains a challenge. In the future, we will consider designing more flexible community sampling methods and message-passing framework for different data types.

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

# A  APPENDIX

## A.1  MORE RELATED WORKS

**Higher-Order Representation learning.** In computer vision, it is a common approach to divide the whole image into multiple local patches. Vision Transformers (ViTs) Dosovitskiy et al. (2020) then generate the image representation by aggregating high-level representations from these patches rather than individual pixels. Following the transformer architecture, Han et al. Han et al. (2021) further subdivide each local patch into smaller patches. This innovative approach enables the model to capture more detailed representations, thus enhancing feature representations. The high-order, or high-level representations derived from local patches, which often share similar content, play a critical role in learning visual representations. In the graph domain, several studies Feng et al. (2019); Wang et al. (2022) also consider encoding higher-order correlations for graph representation learning. Typically, the hypergraph structure with a series of hyperedges is introduced to model the complex higher-order relationship. Within the context of GTs, some recent studies Gao et al. (2022a); Zhao et al. (2023) have attempted to extract substructures, treat them as patches, and utilize the substructural representations for graph classification tasks. As graphs continue to grow rapidly in size, the relationships among nodes become increasingly complex. Therefore, exploring and exploiting higher-order representations is essential for graph representation learning.

**Virtual Node in Message-Passing.** The introduction of a virtual node expands the graph by adding an extra node that facilitates information exchange among all pairs of nodes. Its effectiveness in improving performance has been observed in various tasks Hu et al. (2021b). Recently, there has been a significant focus on studying its theoretical properties. Hwang et al. Hwang et al. (2022) analyzed the virtual node's role in the context of link prediction. They found that virtual nodes can help to add expressiveness of the learned link representation and decrease under-reaching and over-smoothing. Cai et al. Cai et al. (2023) demonstrated the power of message-passing with a virtual node, showing that it can approximate an arbitrary self-attention layer within GTs. While the function of virtual node as READOUT has been explored in existing GNNs, the community nodes in our HOGT have a slightly different function. In addition to aggregate information like the READOUT, they act as bridges connecting the entire graph to propagate long-range dependent information, while also saving computational costs, as the number of communities is significantly smaller than the number of graph nodes.

## A.2  COMPLEXITY ANALYSIS OF HOGT

We analyze the complexity of HOGT. The computational complexity of the first step Graph Node-to-community Node is $\mathcal{O}(mN)$. Since $m$ is the number of community and usually much smaller than

the number of graph nodes $N$, the computational complexity can be simplified as $\mathcal{O}(N)$. Moreover, the computational complexity of the second step community Node-to-community Node is $\mathcal{O}\left(m^2\right)$, it is a self-attention. The final step community Node-to-community Node is $\mathcal{O}\left(N\right)$. Therefore, the overall complexity of HOGT is $\mathcal{O}(m^2 + N)$.

## A.3 THE COMMUNITY SAMPLING METHODS

**Random walk sampling.** To preserve the graph structural information as well as local or long-range connectivity, random walk sampling is a simple but effective approach. We consider a regular random walk sampler with $m$ root nodes selected uniformly at random and each walker goes $k$ hops. As such, we can obtain the communities $\left\{\tilde{\mathcal{V}}_1, \ldots, \tilde{\mathcal{V}}_m\right\}$. Each community $\tilde{\mathcal{V}}_i$ has $k+1$ nodes which are $k$-hop neighbours.

**Spectral clustering.** Spectral clustering methods segment the graph by minimum cuts such that the number of within-cluster links is much higher than between-cluster links in order to better capture good community structure. However, these spectral clustering methods can just obtain non-overlapping clusters. As we aim to achieve more communication between communities, we extend each cluster with its 1-hop neighbourhood He et al. (2023). Thus, we can obtain $m$ communities $\left\{\tilde{\mathcal{V}}_1, \ldots, \tilde{\mathcal{V}}_m\right\}$, where $\tilde{\mathcal{V}}_i \leftarrow \tilde{\mathcal{V}}_i \cup \left\{\mathcal{N}_1(j) \mid j \in \tilde{\mathcal{V}}_i\right\}$.

**Learnable sampling.** For regular graphs, we explore a learnable method that employs reinforcement learning to determine the optimal number of clusters.

Given graph $\mathcal{G} = (\mathcal{V}, \mathcal{E})$ with node set $\mathcal{V}$ and edge set $\mathcal{E}$. Suppose there are $N$ nodes in $\mathcal{V}$. The graph topology is presented by the adjacency matrix $\mathbf{A}$. First, we learn a GNN-based encoder:

$$\mathbf{H}_\ell = \text{GNN}\left(\mathbf{A}, \mathbf{H}_{\ell-1}\right), \ell = 1, \ldots, L, \tag{9}$$

and obtain the representation of $N$ nodes $\mathbf{H} = \left[\boldsymbol{h}_1^\top, \ldots, \boldsymbol{h}_N^\top\right]^\top \in \mathbb{R}^{N \times d}$. Then, we employ a trainable projection vector $\mathbf{p}$ to project all node features to 1D. Given node $v_i$ with feature $\boldsymbol{h}_i$, the scale projection of $x_i$ on $\mathbf{p}$ is $y_i = \boldsymbol{h}_i\mathbf{p}/\|\mathbf{p}\|$. Here, $y_i$ measures how much information of node $v_i$ can be retained when projected to the direction of $\mathbf{p}$. After that, we adopt top-$k$ sampling to select $kN$ nodes, here $k \in (0, 1]$. For each selected node $i$, we generate a community $\tilde{\mathcal{V}}_i$ with its neighbors.

To find the optimal $k$ in top-$k$ sampling, we present a reinforcement learning (RL) algorithm to update the sampling ratio $k$ adaptively. We model the updating process of $k$ as a finite horizon Markov Decision Process (MDP). Formally, the state, action, transition, reward and termination of the MDP are defined as follows:

*State.* The state $s_e$ at epoch $e$ is represented by the indices of selected nodes with pooling ratio $k$:

*Action.* RL agent updates $k$ by taking action $a_e$ based on reward. We define the action $a_e$ as add or minus a fixed value $\Delta k \in [0, 1]$ from $k$.

*Transition.* After updating $k$, we use top-$k$ sampling to select a new set of nodes and corresponding communities in the next epoch.

*Reward.* Due to the black-box nature of GTs, it is hard to sense its state and cumulative reward. So we define a discrete reward function $\text{reward}\left(s_e, a_e\right)$ for each $s_e$ at $a_e$ directly based on the classification results:

$$\text{reward}\left(s_e, a_e\right) = \begin{cases} +1, & \text{if } acc_e > acc_{e-1} \\ 0, & \text{if } acc_e = acc_{e-1} \\ -1, & \text{if } acc_e < acc_{e-1}, \end{cases} \tag{10}$$

where $acc_e$ is the classification accuracy at epoch $e$. Eq. (4) indicates if the classification accuracy with $a_e$ is higher than the previous epoch, the reward for $a_e$ is positive, and vice versa.

*Termination.* If the change of $k$ among 10 consecutive epochs is no more than $\Delta k$, the RL algorithm will stop and $k$ will remain fixed during the next training process. This means that RL finds the

optimal threshold that can retain the most striking nodes. The terminal condition is formulated as:

$$\text{Range } (\{k_{e-10}, \cdots, k_e\}) \leq \Delta k. \tag{11}$$

We adopt Q-learning Watkins & Dayan (1992); Sun et al. (2021) to learn the MDP. Q-learning is an off-policy RL algorithm that seeks to find the best action to take given the current state. It fits the Bellman optimality equation as follows:

$$Q^* (s_e, a_e) = \text{reward } (s_e, a_e) + \gamma \arg\max_{a'} Q^* (s_{e+1}, a'), \tag{12}$$

where $\gamma \in [0, 1]$ is a discount factor of future reward. We adopt a $\varepsilon$-greedy policy with an explore probability $\varepsilon$:

$$\pi (a_e \mid s_e; Q^*) = \left\{ \begin{array}{ll} \text{random action,} & \text{w.p. } \varepsilon \\ \arg\max\limits_{a_e} Q^* (s_e, a), & \text{otherwise} \end{array} \right. \tag{13}$$

This means that the RL agent explores new states by selecting an action at random with probability $\varepsilon$ instead of selecting actions based on the max future reward. We train the RL agent and node classification model jointly in an end-to-end manner.

A.4 CONNECTION BETWEEN COMMUNITY NODE AND HYPEREDGE

We analyze the role of community nodes in capturing the high-order representation in HOGT versus the function of hyperedges in hypergraph convolutional networks.

**Encode complex relationship.** To encode the high-order correlations in the complicated graph, in hypergraph convolutional networks (HGCN), the hyperedges are introduced to connect multiple nodes. In this work, we introduce a community node for each community which contains multiple nodes sharing similar properties (semantic or information). Like the hyperedge, the community node connects with every node in its community.

**High-Order Message-Passing.** Following the message-passing scheme, HGCN first propagates and aggregates information along hyperedge $e^h$ to obtain the hyperedge presentation $\boldsymbol{a}_{e^h}$, then updates the node representation by aggregating the hyperedge representations. Formally, the layer-wise message-passing is defined as:

$$\boldsymbol{a}_{e^h}^{(k)} = \text{Aggregate}^{(k)} \left( \left\{ \boldsymbol{z}_u^{(k-1)} : u \in e^h \right\} \right), \boldsymbol{z}_v^{(k)} = \text{Update}^{(k)} \left( \left\{ \boldsymbol{a}_{e^h}^{(k)} : v \in e^h \right\} \right), \tag{14}$$

where $\boldsymbol{z}_v^{(k)}$ is the feature vector of node $v$ at the $k^{th}$ layer. The hypergraph-based convomutional networks design $\text{Aggregate}^{(k)}(\cdot)$ and $\text{Combine}^{(k)}(\cdot)$ operations based on hypergraph structure.

For example, in a spectral-based hypergraph convolutional network, the convolutional operation is defined as:

$$\boldsymbol{\Delta} = \mathbf{D}_v^{-1/2} \mathbf{S} \mathbf{W} \mathbf{D}_e^{-1} \mathbf{S}^\top \mathbf{D}_v^{-1/2}, \boldsymbol{h}^{(k)} = \sigma \left( \boldsymbol{\Delta} \boldsymbol{Z}^{(k-1)} \boldsymbol{\Theta}^{(k)} \right), \tag{15}$$

where the diagonal matrices $\mathbf{D}_v$ and $\mathbf{D}_e$ denote the vertex and hyperedge degrees, respectively. $\mathbf{W}$ indicate the relationship of hyperedges, the incidence matrix $\mathbf{S}$ denote the correlations of nodes and hyperedges with $S(v, e) = \left\{ \begin{array}{ll} 1, & \text{if } v \in e \\ 0, & \text{if } v \notin e \end{array} \right.$, $\Theta^k$ is the weights of $k^{th}$ layer. Based on the hyperedge operation, we can refine the message-passing in Eq. 15 into three steps: node-to-hyperedge, hyperedge-to-hyperedge, hyperedge-to-node with the approximate presentation:

$$\boldsymbol{a}_{e^h}^{(k)} = \mathbf{S}^\top \boldsymbol{z}^{(k-1)}, \boldsymbol{a}_{e^h}^{(k)} = \mathbf{W} \boldsymbol{a}_{e^h}^{(k)}, \boldsymbol{z}^{(k)} = \mathbf{S} \boldsymbol{a}_{e^h}^{(k)}. \tag{16}$$

We can see that the three-step message-passing in HGCN is equivalent to the three-step operation in HOGT. In HGCN, the relationship of hyperedges usually can be ignored, i.e., $\mathbf{W} = \mathbf{I}$. In HOGT, the framework can also be simplified to two steps without Community Node-to-Community Node. From a high level, graph convolutional neural networks can be viewed as special cases of hypergraph convolutional networks. In comparison, our proposed HOGT framework can be simplified to other existing GT models.

A.5 PROOF

**Proof of Proposition 4.1** Here, we briefly show how the approximation error can be bounded in Proposition 4.1. The complete proof can be found in Cai et al. (2023).

*Proof.* We first make the following assumptions on the feature space $\mathcal{X} \subset \mathbb{R}^{n \times d}$ and the regularity of layer **L**.

**Assumption 1.** $\forall i \in [n]$, $\boldsymbol{x}_i \in \mathcal{X}_i$, $\|\boldsymbol{x}_i\| < C_1$. This implies $\mathcal{X}$ is compact.

**Assumption 2.** $\|\boldsymbol{W}_Q\| < C_2$, $\|\boldsymbol{W}_K\| < C_2$, $\|\boldsymbol{W}_V\| < C_2$ for target layer **L**. Combined with Assumption 1 on $\mathcal{X}$, this means the unnormalized attention $\alpha'(\boldsymbol{x}_i, \boldsymbol{x}_j) = \boldsymbol{x}_i^T \boldsymbol{W}_Q (\boldsymbol{W}_K)^T \boldsymbol{x}_j$ is both upper and lower bounded, which further implies $\sum_j e^{\alpha'(\boldsymbol{x}_i, \boldsymbol{x}_j)}$ be both upper bounded and lower bounded.

Under Assumptions 1 and 2, MPNN+CN of $\mathcal{O}(1)$ width and $\mathcal{O}(1)$ depth can approximate $\boldsymbol{L}_{\text{Performer}}$ and $\boldsymbol{L}_{\text{Linear-Transformer}}$ arbitrarily well. Specifically, $\phi$ can be approximated arbitrarily well by MLP with $\mathcal{O}(1)$ width and $\mathcal{O}(1)$ depth Cybenko (1989), $\phi(\boldsymbol{q}_i)$, $\sum_{j=1}^n \phi(\boldsymbol{k}_j) \otimes \boldsymbol{v}_j$ lies in a compact domain ($n$ is fixed) as $\phi$ is continuous, $\phi(\boldsymbol{q}_i)^T \sum_{k=1}^n \phi(\boldsymbol{k}_k)$ is uniformly lower bounded by a positive number for any node features in $\mathcal{X}$. In Proposition 4.1, we consider Linear Transformer for convenience. $\qquad\square$

**Proof of Theorem 4.1** The "full" self-attention can be approximated following: 1) Message-Passing Neural Networks with community nodes (MPNN+CN) can act as the self-attention layer, and 2) Under our three-step message-passing framework, the combination of MPNN+CN with the self-attention can achieve the approximated full self-attention in graph. While point 1) has been validated in Proposition 1, we mainly demonstrated point 2).

*Proof.* In the process of Graph Node-to-Community Node (**G2C**-**MP**), the message-passing in a community is powerful to update community node (cn) by aggregate the information fom graph nodes (gn) as:

$$h_i^{(k)} = \tau_{j \in \mathcal{C}(i)} \phi_{\text{gn-cn}}^{(k)} \left( h_i^{(k-1)}, x_j^{(k-1)}, e_{j,i} \right), \tag{17}$$

where $\phi$ is message function, and $\tau$ is aggregation function, $\mathcal{C}(i)$ is the graph nodes in the community $i$. Based on **Proposition 4.1**, the message-passing with a new introduced node that connected to every nodes in the community can be approximated by the following aggregation function $\tau$:

$$h_i^{(k)} = \tau_{j \in \mathcal{C}(i)} \phi_{\textbf{G2C}-\textbf{MP}}^{(k)} \left( \cdot, \{\boldsymbol{x}_i\}_i \right) = \left[ \sum_{j=1}^{|\mathcal{C}|} \phi(\boldsymbol{k}_j), \ f \left( \sum_{j=1}^{|\mathcal{C}|} \phi(\boldsymbol{k}_j) \otimes \boldsymbol{v}_j \right) \right], \tag{18}$$

where $f(\cdot)$ flattens a 2D matrix to a 1D vector in raster order, $\boldsymbol{k}_j = \boldsymbol{W}_K^{(k)} \boldsymbol{x}_j^{(k)}$, and $\boldsymbol{v}_j = \boldsymbol{W}_V^{(k)} \boldsymbol{x}_j^{(k)}$.

Then, in the process of Community Node-to-Community Node (**C2C**-**ATTN**), a self-attention mechanism ($\gamma_{\textbf{C2C}-\textbf{ATTN}}$) is adopted to propagate information between any two community nodes. The updated community nodes can be represented as:

$$\overline{h}_i^k = \gamma_{\textbf{C2C}-\textbf{ATTN}} \left( \left[ \sum_{j=1}^m \phi(\boldsymbol{k}_j), \ f \left( \sum_{j=1}^m \phi(\boldsymbol{k}_j) \otimes \boldsymbol{v}_j \right) \right] \right), \tag{19}$$

where $m$ is the number of communities, $\boldsymbol{k}_j = \boldsymbol{W}_K^{(k)} h_j^{(k)}$, and $\boldsymbol{v}_j = \boldsymbol{W}_V^{(k)} h_j^{(k)}$.

Finally, the updated community node sends its message back to graph nodes in its community. Each graph node $v_i$ applies the update function $\gamma_{\text{gn}}$:

$$x_i^{(k)} = \gamma_{\text{gn}}^{(k)} \left( x_i^{(k-1)}, \tau_{j \in \mathcal{V}_{(i)}} \phi_{\text{cn-gn}}^{(k)} \left( x_i^{(k-1)}, \overline{h}_j^{(k-1)}, e_{j,i} \right) \right), \tag{20}$$

where $\mathcal{V}_{(i)}$ is the the community set of graph node $i$. Based on **Proposition 4.1**, the message-passing in the step Community Node-to-Graph Node (**C2G-MP**) can be formulated as:

$$x_i^{(k)} = \gamma_{\textbf{\textit{C2G}-MP}} \left( \boldsymbol{x}_i, \left[ \sum_{j=1}^{\lceil |\mathcal{V}_{(i)}| \rceil} \phi\left(\boldsymbol{k}_j\right), f\left( \sum_{j=1}^{\lceil |\mathcal{V}_{(i)}| \rceil} \phi\left(\boldsymbol{k}_j\right) \otimes \boldsymbol{v}_j \right) \right] \right) \tag{21}$$

where $\boldsymbol{k}_j = \boldsymbol{W}_K^{(k)} \overline{h}_j^{(k)}$, and $\boldsymbol{v}_j = \boldsymbol{W}_V^{(k)} \overline{h}_j^{(k)}$.

Following the three-step architecture, the information of a graph node can be propagated to any other nodes by the community nodes as the bridges. And the representations of graph nodes can be approximated as:

$$x_i^k = \frac{\left( \phi\left(\boldsymbol{q}_i\right) \sum_{j=1}^{n} \phi\left(\boldsymbol{k}_j\right) \otimes \boldsymbol{v}_j \right)^T}{\phi\left(\boldsymbol{q}_i\right)^T \sum_{k=1}^{n} \phi\left(\boldsymbol{k}_k\right)}, \tag{22}$$

where $n$ is the number of graph nodes, $\boldsymbol{q}_i = \boldsymbol{W}_Q^{(k)} x_i^{(k)}$, $\boldsymbol{k}_j = \boldsymbol{W}_K^{(k)} x_j^{(k)}$, and $\boldsymbol{v}_j = \boldsymbol{W}_V^{(k)} x_j^{(k)}$. Therefore, the combination of Message-Passing with a new node followed by a self-attention followed by another Message-Passing can approximate self-attention arbitrarily well.

$\square$

## A.6 EXPERIMENTAL PART

**Settings.** For Cora, Citeseer, and Pubmed datasets, we follow the same experimental procedure, such as features and data splits in Pei et al. (2020). For heterophilic graph datasets (Cornell, Texas, Wisconsin, and Actor), we adopt the same dataset splits used by Zhu et al. (2020). For roman-empire and amazon-ratings, we follow the settings in Platonov et al. (2023). For hypergraphs, we adopt the same setting as Yadati et al. (2019); Chien et al. (2021a). For other datasets, we randomly split them into 60%/20%/20% as training/validation/test sets following Zhang et al. (2022a); Liu et al. (2023). The dataset obgn-arxiv can be downloaded from Open Graph Benchmark (OGB) Hu et al. (2020a) [1], hypergraph datasets from [2], roman-empire and amazon-ratings from [3], all the other graph datasets from PyTorch Geometric (PyG) Fey & Lenssen (2019) [4]

For the general sampling methods-random walk Zeng et al. (2019) and spectral clustering Chiang et al. (2019), we set the number of communities to $1$ (the whole graph as a community) and $1\%, 10\%, 20\%, 50\%$ of the number of nodes in the graph. For the proposed learnable sampling method, the optimal number of communities can be actively learned. The training utilizes Adam optimizer Kingma & Ba (2014) for GNN methods, while Adamw is adopted for all Graph Transformer-based models. Each method runs for 200 epochs on all datasets, with the test accuracy reported based on the epoch that achieves the highest validation accuracy. We set 3 layers HOGT for ogbn-arxiv, 5 layers for roman-empire and amazon-ratings, and 2 layers for other datasets. We search model hyper-parameters including walk length of random walk, hidden dimension, and dropout. The results of HOGT are averaged over 10 runs with random weight initializations. Furthermore, all the experiments are conducted on a Linux server equipped with NVIDIA A100.

## A.7 INITIALIZING COMMUNITY NODES

Community nodes are crucial for the proposed HOGT method. In this section, we analyze the strategies for initializing these nodes.

The community nodes can be initialized either with zero vectors or random values. Our experiments show that both approaches lead to similar final performance after 200 epochs, suggesting that the choice of initialization has minimal impact on the final outcomes. In our experiments, we use random initialization and set the community node dimensionality to match the original node features.

---

[1]OGB: https://ogb.stanford.edu/docs/nodeprop/#ogbn-arxiv

[2]DHG: https://deephypergraph.readthedocs.io/en/latest/index.html

[3]DGL: https://docs.dgl.ai/

[4]PyG: https://github.com/pyg-team/pytorch_geometric

Table 10: Node classification results on hypergraph datasets (mean accuracy (%) and standard deviation over 5 different runs). The complexity of information propagation can be found for different models. The number of nodes, edges, and communities are $|E|$, $N$, and $m$, respectively.

| Model | Coauthor-Cora | Coauthor-DBLP | News20 | Complexity |
|---|---|---|---|---|
| GCN | 64.42±0.68 | 81.35±0.18 | 76.82±0.48 | $\mathcal{O}(|E|)$ |
| HGNN | 61.18±0.62 | 82.66±1.05 | 81.06±1.03 | $\mathcal{O}(N^2)$ |
| HGNN+ | 60.40±0.77 | 82.86±0.85 | 81.24±0.75 | $\mathcal{O}(N^2)$ |
| HOGT (ours) | **68.82±1.34** | **85.82 ±0.70** | **81.32±0.80** | $\mathcal{O}(m^2 + N)$ |

Although random or zero initialization is effective, an alternative strategy—such as using max or mean pooling of the features of graph nodes within a community to initialize the community node features—could potentially accelerate convergence by providing a more informed starting point for community node embeddings. However, this method introduces additional computational overhead, which we have intentionally avoided in our current implementation to maintain efficiency.

## A.8 DATASET STATISTIC.

Table 7: Statistics of graph benchmark datasets.

| | Cora | Citeseer | Pubmed | ogbn-arxiv | Cornell | Texas | Wisconsin | Actor | roman-empire | amazon-ratings |
|---|---|---|---|---|---|---|---|---|---|---|
| **# Nodes** | 2,708 | 3,327 | 19,717 | 169,343 | 183 | 183 | 251 | 7,600 | 22,662 | 24,492 |
| **# Edges** | 5,429 | 4,732 | 44,338 | 1,166,343 | 280 | 195 | 466 | 26,752 | 32,927 | 93,050 |
| **Homo.** | 0.83 | 0.72 | 0.79 | 0.63 | 0.30 | 0.11 | 0.21 | 0.22 | 0.05 | 0.38 |

Table 8: Statistics of hypergraph benchmark datasets.

| | Coauthorship-Cora | Coathorship-DBLP | News20 |
|---|---|---|---|
| **# Nodes** | 2,708 | 41,302 | 16,342 |
| **# Hyperedges** | 1,072 | 22,363 | 100 |
| **# Classes** | 7 | 6 | 4 |

## A.9 MORE RESULTS AND EXPLANATIONS.

Table 9: The p-values of the t-test between the performances of different methods.

| Model | Cornell | Actor | roman-empire |
|---|---|---|---|
| Mixhop/HGT | 0.026 | 5.67e-07 | 8.36e-13 |
| GPRGNN/HGT | 0.016 | 1.36e-06 | 7.82e-27 |
| Gapformer/HGT | 0.037 | 0.006 | 0.0009 |

**Performance on Hypergraphs.** Theoretically, both hypergraph convolutional networks (HGCN) and our HOGT can learn high-order correlations in complex datasets. Here, based on hypergraph structure, we generate a community for each hyperedge. According to the results in Table 10, HOGT achieves better performance than popular hypergraph methods HGNN Feng et al. (2019) and HGNN+ Gao et al. (2022b) across all hypergraph datasets. Compared to traditional HGCN methods, HOGT can propagate higher-order information more flexibly based on attention architecture.

**The fail of positional encoding on heterophilic datasets.** To better explain this phenomenon, we first show how positional encoding is related to the theoretical properties of GTs, e.g., their expressive power in capturing graph structure.

The implementation of PE, i.e., concatenated with input features, tends to influence the attention scores, producing an attention bias. Considering that $\mathbf{Q} \in \mathbb{R}^{n \times d}$, $\mathbf{K} \in \mathbb{R}^{n \times d}$, and $\mathbf{P} \in \mathbb{R}^{n \times d'}$

represent the query, key, and PE vectors, respectively, the attention score $\mathbf{S} \in \mathbb{R}^{n \times n}$ is calculated as:

$$\mathbf{S} = \mathbf{Q}\mathbf{K}^\top. \tag{23}$$

After concatenating the PE vector, the refined attention score $\mathbf{S}'$ is calculated as:

$$
\begin{aligned}
\mathbf{S}' &= [\mathbf{Q}, \mathbf{P}] \times [\mathbf{K}, \mathbf{P}]^\top \\
&= [\mathbf{Q}, \mathbf{P}] \times \left[ \begin{array}{c} \mathbf{K}^\top \\ \mathbf{P}^\top \end{array} \right] \\
&= \mathbf{Q}\mathbf{K}^\top + \mathbf{P}\mathbf{P}^\top,
\end{aligned} \tag{24}
$$

where $[\mathbf{Q}, \mathbf{P}]$ denotes the concatenation of the query vector $\mathbf{Q}$ with the PE vector, and $[\mathbf{K}, \mathbf{P}]$ denotes the concatenation of the key vector $\mathbf{K}$ with the PE vector. The $\mathbf{P}\mathbf{P}^\top$ term can be interpreted as an attention bias.

Inappropriate positional encoding can affect the attention matrix, leading to a negative impact on performance. Muller et al. Müller et al. (2023) clarified that no clear expressivity hierarchy exists for the popular positional or structural encodings, including Laplacian PE and RandomWalk PE. In other words, the critical aspects of existing PEs in GT haven't been demonstrated theoretically and empirically.

From Table 4 in the paper, the performance gap is minor with or without positional encoding methods on homophilic datasets (Cora and Citeseer). Without positional encoding, HGT demonstrates a better performance on heterophilic datasets, such as Cornell, Texas, and Wisconsin. This implies that existing positional encoding methods cannot accurately capture the structural information from heterophilic datasets, which is consistent with the above analysis. This motivates researchers to design more suitable positional encoding methods for different datasets or explore alternative approaches to encode the graph structural information like our HGT framework.

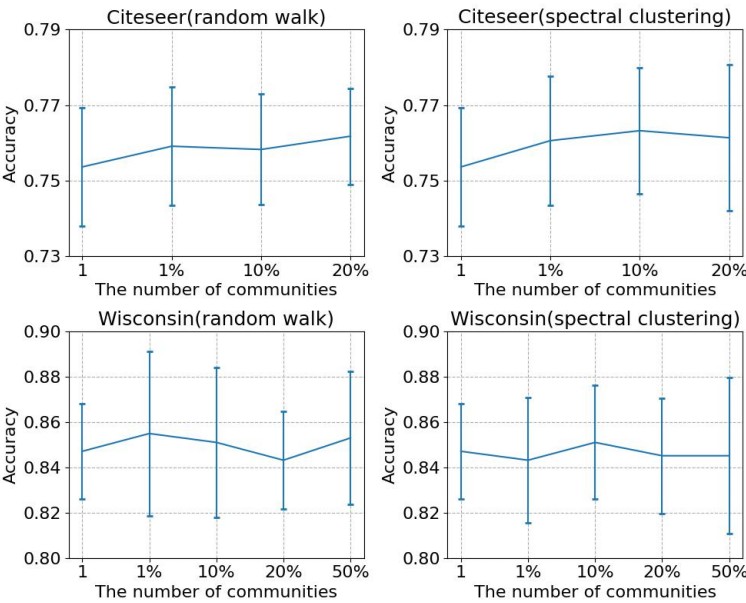

Figure 2: The ablation study on the number of communities. We set the number of communities to 1 (the whole graph as a community) and $1\%, 10\%, 20\%, 50\%$ of the number of graph nodes.

**Effect of the Number of Community.**    We analyze the effect of the number of communities with the two unlearnable sampling methods for HOGT. From the results in Figure 2, we see that increasing the number of communities in the early stage can enhance the performance of HOGT (randomwalk) on Cora. This is because HOGT encodes more local higher-order information with more communities extracted by random walk. As the number of communities increases, we can

Table 11: Abalation study of different components of HOGT on different datasets (mean accuracy (%) and standard deviation over 10 different runs).

| Community Sampling | Model | Cora | Citeseer | Cornell | Texas | Wisconsin |
|---|---|---|---|---|---|---|
| Random Walk | HOGT(w/o *C2C*-**ATTN**)) | 87.73±0.96 | 74.94±1.64 | 77.57±3.21 | 80.54±3.59 | 85.89±2.60 |
| | HOGT | 88.11±1.05 | 76.74±1.47 | 76.49±2.72 | 80.00±4.22 | 87.25±2.67 |
| Random Walk | HOGT(w/o local) | 83.04±1.48 | 74.47±2.10 | 76.49±2.72 | 82.70±4.86 | 83.44±1.87 |
| | HOGT(w local) | 88.11±1.05 | 76.74±1.47 | 70.27±2.34 | 74.90±2.78 | 78.19±2.67 |

observe a decreasing trend followed by an increase for HOGT with the spectral clustering method on Cora. This illustrates that there likely exist some important substructures in the graph. We also note the stable performance of HOGT on Wisconsin with different numbers of communities for both methods. While Wisconsin is a small-scale dataset, the global information can be well encoded by introducing a community.

The RL-based sampling method adaptively learns the optimal number of communities, eliminating the need to predefine this hyperparameter. This approach adds flexibility to HOGT and ensures robust performance without requiring extensive manual tuning of the number of communities.

**Effect of Self-Attention Between Communities.**    As we analyzed in Appendix A.4, if dropping out the second step (*C2C*-**ATTN**), in terms of message-passing, HOGT behaves similarly to popular hypergraph-based neural networks. In this case, we are not taking into account the relationships between communities and we can see that in Table 11, HOGT (w/o *C2C*-**ATTN**)) exhibits a performance degradation compared to HOGT on datasets which have complex structure (like more nodes and edges). Without *C2C*-**ATTN**, the node representation is still limited in the local neighbourhood, i.e., community. Propagating information between communities can help the node finally capture the higher-order long-range dependency in the whole graph.

**Effect of Local Information for Different Datasets.**    Given one of the major advantages of Transformer is capturing the long-range dependency in objects, we examine the importance of local information for some of the benchmarks. From Table 11, we note that it can improve the performance if we consider the local neighbours in the third step (*G2V*-**MP**) for Cora and Citeseer as they are small-scale datasets with high **Homo.**. In contrast, it is more beneficial to disregard the original graph connections for Cornell, Texas, and Wisconsin with low **Homo.**.

**Performance on Graph Classification.**    We utilize several commonly-used real-world datasets from TU database Morris et al. (2020) to evaluate the performance of HOGT on graph classification task. **NCI1** consists of 4,110 molecule graphs from TUDataset, which represent two balanced subsets of datasets for chemical compounds screened for activity against non-small cell lung cancer and ovarian cancer cell lines, respectively. **PROTEINS** consists of 1,113 protein graphs from TUDataset, where each graph corresponds to a protein molecule, nodes represent amino acids, and edges capture the interactions between amino acids. From Table 12, we can observe that HOGT can achieve state-of-the-art performance on all datasets. Compared to GT models like GraphGPS, HOGT can encode more comprehensive information in the graph.

Table 12: Experimental results on two datasets (the mean accuracy (**Acc.**) and standard deviation over 10 different runs).

| | NCI1 | PROTEINS |
|---|---|---|
| *GCN-based methods* | | |
| GCN Kipf & Welling (2017) | $79.68_{\pm2.05}$ | $71.7_{\pm4.7}$ |
| GAT Veličković et al. (2018) | $79.88_{\pm0.88}$ | $72.0_{\pm3.3}$ |
| GIN Xu et al. (2019b) | $81.7_{\pm1.7}$ | $73.76_{\pm4.61}$ |
| GatedGCN Li et al. (2016) | $81.17_{\pm0.79}$ | $74.65_{\pm1.13}$ |
| *Graph Transformer-based methods* | | |
| GT Dwivedi & Bresson (2021) | $80.15_{\pm2.04}$ | $73.94_{\pm3.78}$ |
| SAN Kreuzer et al. (2021) | $80.50_{\pm1.30}$ | $74.11_{\pm3.07}$ |
| Graphormer Ying et al. (2021) | $81.44_{\pm0.57}$ | $75.29_{\pm3.10}$ |
| GraphTrans Wu et al. (2021) | $82.60_{\pm1.20}$ | $75.18_{\pm3.36}$ |
| SAT Chen et al. (2022a) | $80.69_{\pm1.55}$ | $73.32_{\pm2.36}$ |
| GraphGPS Rampášek et al. (2022) | $84.21_{\pm2.25}$ | $75.77_{\pm2.19}$ |
| GT(a whole graph as a community) | $\mathbf{84.67}_{\pm\mathbf{1.32}}$ | $\mathbf{76.78}_{\pm\mathbf{1.84}}$ |

**Performance on Link Prediction.** We have conducted experiments on the TEG-DB datasets (Li et al. (2024)) to further evaluate the performance of HOGT on link prediction, specifically on the Goodreads-Children and Goodreads-Crime datasets. These datasets are Textual-Edge Graphs (TEG) with rich textual descriptions for nodes and edges. Models like BERT-Large (Devlin (2018)) and BERT-Base (Devlin (2018)) were used to obtain node and edge embeddings. The results in Table 13 demonstrate that HOGT achieves performance that is either better than or comparable to General-Conv (You et al. (2020)) and GraphTransformer (Yun et al. (2019)). In these experiments, we treated the data in each batch as a community and introduced a community node for each batch, effectively extending the HOGT framework to these diverse domains.

Table 13: Performance Comparison Across Datasets and Methods (w/o edge text indicates that edge embeddings are not used).

| Methods | Goodreads-Children | | | | | | Goodreads-Crime | | | | | |
|---|---|---|---|---|---|---|---|---|---|---|---|---|
| | BERT-Large | | BERT-Base | | w/o edge text | | BERT-Large | | BERT-Base | | w/o edge text | |
| | AUC | F1 | AUC | F1 | AUC | F1 | AUC | F1 | AUC | F1 | AUC | F1 |
| GeneralConv | 0.9810 | 0.9179 | 0.9821 | 0.9187 | **0.9825** | 0.9189 | 0.9772 | 0.9079 | 0.9774 | 0.9077 | 0.9752 | 0.9101 |
| GraphTransformer | 0.9807 | 0.9200 | 0.9811 | 0.9160 | 0.9776 | 0.9066 | 0.9738 | 0.9079 | 0.9737 | 0.9079 | 0.9716 | 0.8983 |
| HOGT | **0.9821** | **0.9216** | **0.9837** | **0.9208** | **0.9825** | **0.9289** | **0.9776** | **0.9110** | **0.9776** | **0.9110** | **0.9768** | **0.9130** |

## A.10 HYPERPARAMETER ANALYSIS

We conducted an analysis of HOGT's sensitivity to various hyperparameters, including walk length, hidden dimension, dropout rate, and optimizer. The results are summarized in Table 14. From the findings, we observe that HOGT, when using the random walk sampling method, demonstrates low sensitivity to walk length and dropout rate. However, on the Cora dataset, the model shows higher sensitivity to the hidden dimension and optimizer, indicating their importance in influencing performance. In our experiments, we adopted AdamW as the optimizer for HOGT and other GT models.

Table 14: The performances of HOGT with different hyperparameters.

| **Hyperparameters** | | Cora | Citeseer | Pubmed |
|---|---|---|---|---|
| Hidden dimension | 128 | 85.45 | 76.63 | 88.42 |
| | 256 | 88.13 | 76.76 | 89.20 |
| Dropout | 0 | 87.73 | 76.98 | 88.40 |
| | 0.2 | 86.92 | 76.76 | 88.40 |
| | 0.5 | 87.05 | 76.93 | 89.02 |
| walk length | 3 | 86.59 | 76.68 | 88.39 |
| | 5 | 87.59 | 76.73 | 88.41 |
| | 10 | 87.45 | 76.98 | 88.45 |
| Optimizer | Adam | 86.91 | 76.08 | 88.24 |
| | Adamw | 88.11 | 76.74 | 89.20 |

## A.11 ROBUSTNESS ANALYSIS

To further evaluate the robustness of the proposed HOGT in handling graphs with sparse structure, we conducted additional experiments by randomly removing 10% and 20% of the edges from Citeseer and Pubmed. The results in Table 15, demonstrate that HOGT maintains strong performance even under these conditions. This highlights the robustness of HOGT in processing graphs with irregular or sparse structures.

Table 15: The performance of HOGT (randomwalk) with sparse structure on Citeseer and Pubmed. The edge ratio means the reserving ratio of original edges.

| Method | Edge Ratio | Citeseer | Pubmed |
|--------|-----------|----------|--------|
|        | 80%       | 74.52    | 85.96  |
| HOGT   | 90%       | 75.07    | 86.71  |
|        | 100%      | 76.74    | 89.20  |

