# OpenReview forum: "HOGT: High-Order Graph Transformers"
_ICLR.cc/2025/Conference — Submitted to ICLR 2025_

### Official Review · Reviewer_RHuk · 2024-10-17

**Soundness:** 3
**Presentation:** 3
**Contribution:** 3
**Rating:** 6
**Confidence:** 3

**Summary:**

The paper presents a unique approach to graph learning by integrating high-order information propagation within the transformer architecture. The paper empirically shows that HOGT achieves competitive results on node and graph classification tasks, especially on heterophilic datasets.

**Strengths:**

The idea of using a learnable community sampling method with reinforcement learning for graph representation is novel. It combines the advantages of community detection and adaptive sampling.By addressing the limitations of existing graph transformers in terms of capturing topological information and scalability, this work contributes to the advancement of the field and opens up new research directions for further exploration.

**Weaknesses:**

Analysis on the sensitivity of the HOGT model's performance to its hyperparameters such as walk length, hidden dimension, and dropout.

Further exploration of the sampling method's performance in graphs with irregular or sparse structures would enhance the understanding of the model's robustness.

A more detailed comparison of HOGT's computational complexity, including training time and memory usage, with other state-of-the-art models is needed.

**Questions:**

Could the authors provide more insights into how HOGT scales with graph size, especially in terms of memory usage and training efficiency?


Can the authors elaborate on the theoretical analysis of the model's expressiveness and how it relates to the approximation of global attention?

How sensitive is HOGT to its hyperparameters, particularly the number of communities and the reinforcement learning-based sampling method?

Could the authors discuss how HOGT captures long-term dependencies in the graph and compare this with other methods that focus on long-range interactions?

How sensitive is the performance of HOGT to changes in hyperparameters such as the hidden dimension and dropout rate?
Have the authors experimented with different optimization algorithms for hyperparameter tuning, and if so, what were the results?

---

> ### Author Response · Authors · 2024-11-20
> **Response to Reviewer RHuk**
>
> Dear Reviewer,
>
> Thanks for reviewing our paper and the valuable comments. Please find our point-by-point response to your concerns below.
>
> **Q1.**
> Analysis on the sensitivity of the HOGT model's performance to its hyperparameters such as walk length, hidden dimension, and dropout. Have the authors experimented with different optimization algorithms for hyperparameter tuning, and if so, what were the results?
>
> **A1.**
> We conducted an analysis of HOGT's sensitivity to various hyperparameters, including walk length, hidden dimension, dropout, and optimizer. The results are summarized in the following table. From the findings, we observe that HOGT, when using the random walk sampling method, demonstrates low sensitivity to walk length and dropout. However, on the Cora dataset, the model shows higher sensitivity to the hidden dimension and optimizer, indicating its importance in influencing performance. In our experiments, AdamW is adopted for HOGT and other GT models.
>
> **Table: The performances of HOGT with different hyperparameters.**
>
> | **Hyperparameters**   | **Cora** | **Citeseer** | **Pubmed** |
> |-----------------------|----------|--------------|------------|
> | **Hidden dimension**   |          |              |            |
> | 128                   | 85.45    | 76.63        | 88.42      |
> | 256                   | 88.13    | 76.76        | 89.20      |
> | **Dropout**            |          |              |            |
> | 0                     | 87.73    | 76.98        | 88.40      |
> | 0.2                   | 86.92    | 76.76        | 88.40      |
> | 0.5                   | 87.05    | 76.93        | 89.02      |
> | **Walk length**        |          |              |            |
> | 3                     | 86.59    | 76.68        | 88.39      |
> | 5                     | 87.59    | 76.73        | 88.41      |
> | 10                    | 87.45    | 76.98        | 88.45      |
> | **Optimizer**          |          |              |            |
> | Adam                  | 86.91    | 76.08        | 88.24      |
> | AdamW                 | 88.11    | 76.74        | 89.20      |
>
> **Q2.**
> Further exploration of the sampling method's performance in graphs with irregular or sparse structures would enhance the understanding of the model's robustness.
>
> **A2.**
> Citeseer and Pubmed can be considered sparse graphs, with node-to-edge ratios of 1.4 and 2.2, respectively. To further evaluate the robustness of the proposed HOGT in handling graphs with fewer edges, we conducted additional experiments by randomly removing 10\% and 20\% of the edges from Citeseer and Pubmed. The results, presented in the following table, demonstrate that HOGT, when using the random walk sampling method, maintains strong performance even under these conditions. This highlights the robustness of HOGT in processing graphs with irregular or sparse structures.
>
> **Table: The performance of HOGT (randomwalk) with sparse structure on Citeseer and Pubmed. The edge ratio means the reserving ratio of original edges.**
>
> | **Method** | **Edge Ratio** | **Citeseer** | **Pubmed** |
> |------------|-----------------|--------------|------------|
> | HOGT       | 80%             | 74.52        | 85.96      |
> |            | 90%             | 75.07        | 86.71      |
> |            | 100%            | 76.74        | 89.20      |

---

> ### Author Response · Authors · 2024-11-20
>
> Thank you very much for your patience! Here is the rest part.
>
> **Q3.**
> A more detailed comparison of HOGT's computational complexity, including training time and memory usage, with other state-of-the-art models is needed.
>
> **A3.**
> The following table provides a detailed comparison of the training time per epoch, inference time, and GPU memory usage for popular GNN methods including GAT and APPNP, and GT models including Graphormer, LiteGT, polynormer, and HOGT on the Cora and ogbn-proteins datasets. Since the standard practice for model training on these datasets involves a fixed number of epochs, we report the training time per epoch to effectively compare training efficiency.
>
> The results show that HOGT is orders of magnitude faster than Graphormer and polynormer, which require quadratic complexity for global attention. Additionally, HOGT significantly reduces memory consumption due to its efficient implementation of global attention with $\mathcal{O}(N)$ complexity. Compared to GAT, APPNP, and SGFormer, HOGT strikes a balance between performance and efficiency. This result indicates the effectiveness of HOGT in scaling to large-scale datasets by introducing community nodes and a multi-step message-passing strategy.
>
> **Table: Efficiency comparison of HOGT and graph Transformer competitors w.r.t. training time per epoch, inference time and GPU memory (GB) cost on a A100. The missing results are caused by out-of-memory.**
>
> | **Method**     | **Cora** Tr (ms) | **Cora** Inf (ms) | **Cora** Mem (MB) | **ogbn-proteins** Tr (s) | **ogbn-proteins** Inf (s) | **ogbn-proteins** Mem (MB) |
> |----------------|------------------|-------------------|-------------------|--------------------------|---------------------------|----------------------------|
> | GAT            | 3.18             | 1.68              | 166.35            | -                        | -                         | -                          |
> | APPNP          | 3.32             | 1.49              | 35.57             | -                        | -                         | -                          |
> | Graphormer     | 90.58            | 71.26             | 359.25            | -                        | -                         | -                          |
> | LiteGT         | 15.57            | 5.77              | 227.69            | -                        | -                         | -                          |
> | polynormer     | 218.23           | 5.13              | 264.06            | 1.60                     | 0.127                     | 6429.06                    |
> | SGFormer       | 3.66             | 1.42              | 50.87             | 1.26                     | 0.098                     | 228.19                     |
> | HOGT           | 7.40             | 2.69              | 109.22            | 1.12                     | 0.087                     | 1284.32                    |
>
> **Q4.**
> Can the authors elaborate on the theoretical analysis of the model's expressiveness and how it relates to the approximation of global attention?
>
> **A4.**
> The theoretical foundation for approximating self-attention in HOGT is based on the following: (1) Message-Passing Neural Networks with community nodes (MPNN+CN) can act as a self-attention layer, and (2) under the three-step message-passing framework, the combination of MPNN+CN with self-attention achieves an approximation of full self-attention in graphs.
>
> **For Point (1):**
> The approximation error of self-attention by MPNN+CN can be bounded under the following assumptions. A detailed proof can be found in [1]. Below, we summarize the key assumptions and results:
>
> Assumption 1.
>
> $\forall i \in [n]$, $\boldsymbol{x}_i \in \mathcal{X}_i$, and $|\boldsymbol{x}_i| < C_1$, implying that the feature space $\mathcal{X}$ is compact.
>
> Assumption 2.
>
> $|\boldsymbol{W}_Q| < C_2$, $|\boldsymbol{W}_K| < C_2$, and $|\boldsymbol{W}_V| < C_2$ for the target layer $\mathbf{L}$. Combined with Assumption 1, this ensures that the unnormalized attention $\alpha^{\prime}(\boldsymbol{x}_i, \boldsymbol{x}_j) = \boldsymbol{x}_i^T \boldsymbol{W}_Q (\boldsymbol{W}_K)^T \boldsymbol{x}_j$ is bounded, and $\sum_j e^{\alpha^{\prime}(\boldsymbol{x}_i, \boldsymbol{x}_j)}$ is also upper and lower bounded.
>
> Under these assumptions, MPNN+CN with $\mathcal{O}(1)$ width and $\mathcal{O}(1)$ depth can approximate ${{Performer}}$ and ${\text{Linear-Transformer}}$ arbitrarily well, as shown in [1]. In Proposition 4.1, we consider the Linear Transformer for simplicity.

---

> > ### Author Response · Authors · 2024-11-20
> >
> > Thank you very much for your patience! Here is the rest.
> >
> > **For Point (2):**
> >
> > The three-step message-passing framework enables HOGT to approximate full self-attention as follows:
> >
> > -  Graph Node-to-Community Node (G2C-MP): Message-passing through a newly introduced community node, which is connected to all nodes in the community, approximates self-attention within the community. This is supported by Proposition 4.1 in the paper.
> >
> >  - Community Node-to-Community Node (C2C-ATTN): A self-attention mechanism propagates information among community nodes, enabling communication between communities.
> >
> >  - Community Node-to-Graph Node (C2G-MP): Global information in the community nodes is propagated back to the graph nodes through another MPNN+CN, effectively approximating "full" self-attention across the graph.
> >
> > We focuse primarily on demonstrating Point (2) within the paper, highlighting the interplay between the three-step framework and global attention approximation.
> >
> > [1] Chen Cai, et al. On the connection between MPNN and graph transformer. ICML 2023.
> >
> > **Q5.**
> > How sensitive is HOGT to its hyperparameters, particularly the number of communities and the reinforcement learning-based sampling method?
> >
> > **A5.**
> >
> > We analyzed the sensitivity of HOGT to the number of communities using two unlearnable sampling methods (details in Appendix A.8 of the paper). The results reveal the following trends:
> >
> >    - For **HOGT (random walk)** on the Cora dataset, increasing the number of communities initially improves performance. This improvement occurs because a larger number of communities extracted by random walk allows HOGT to encode more localized higher-order information.
> >    - For **HOGT (spectral clustering)** on Cora, we observe a more complex trend: performance initially decreases with more communities, followed by an improvement. This pattern suggests the presence of critical substructures in the graph that spectral clustering captures effectively when the community structure aligns with these substructures.
> >    - On the Wisconsin dataset, HOGT demonstrates stable performance across different numbers of communities for both methods. Since Wisconsin is a small-scale dataset, introducing a community node effectively encodes the global information without significant dependence on the number of communities.
> >
> > **Reinforcement Learning (RL)-Based Sampling:**
> >    The RL-based sampling method adaptively learns the optimal number of communities, eliminating the need to predefine this hyperparameter. This approach adds flexibility to HOGT and ensures robust performance without requiring extensive manual tuning of the number of communities.
> >
> > These observations demonstrate that while the number of communities can influence HOGT’s performance, the RL-based sampling method provides a practical solution for optimizing this parameter adaptively.

---

> ### Author Response · Authors · 2024-11-20
>
> Thanks again for your patience! Here is the rest.
>
> **Q6.**
> Could the authors discuss how HOGT captures long-term dependencies in the graph and compare this with other methods that focus on long-range interactions?
>
> **A6.**
> In HOGT, long-range dependencies between nodes are effectively captured through the use of community nodes and the three-step message-passing scheme. Community nodes act as intermediaries that aggregate and propagate information from their associated nodes, enabling efficient communication between distant nodes. Specifically, any two graph nodes can interact within fewer than three hops via two community nodes, facilitating the propagation of long-range dependencies with minimal computational overhead.
>
> Compared to HOGT, general Graph Transformer (GT) models address long-range interactions by establishing direct connections between distant nodes through global attention. While this approach is effective for modeling global dependencies, it often incurs substantial computational costs due to the quadratic complexity of global attention. Alternative methods attempt to mitigate this by introducing anchor nodes [1, 2] or supernodes [3, 4], derived from the graph structure to connect distant nodes. However, these methods heavily depend on the initial graph structure, which may inadequately represent and balance critical information in complex or large-scale graphs. More discussion can be found in the Related Work section.
>
> Empirical results on heterophilic and large-scale datasets, which inherently require modeling long-range dependencies, highlight HOGT's effectiveness. Our experiments demonstrate that HOGT consistently outperforms other GT models capable of capturing global information. This performance gain arises from HOGT's ability to efficiently capture and propagate long-range dependencies through its community-node mechanism, while maintaining a lower computational complexity compared to traditional global attention-based methods.
>
> [1] Wenhao Zhu, et al. Anchorgt: Efficient and flexible attention architecture for scalable graph transformers. IJCAI, 2024.
>
> [2] Bo Jiang, et al. Agformer: Efficient graph representation
> with anchor-graph transformer. arXiv preprint arXiv:2305.07521, 2023.
>
> [3] Weirui Kuang, et al. Coarformer: Transformer for
> large graph via graph coarsening, 2022. In URL https://openreview. net/forum, 2021.
>
> [4] Wenhao Zhu, et al. Hierarchical transformer
> for scalable graph learning. IJCAI, 2023.

---

> > ### Author Response · Authors · 2024-11-25
> >
> > Dear Reviewer,
> >
> > Thank you very much for your time and effort in reviewing our paper. Based on your suggestions, we have updated the manuscript (highlighted in blue) with the following additions:
> >
> > Sensitivity analysis of HOGT with different hyperparameters.
> >
> > Robustness analysis of HOGT with sparse graph structures.
> >
> > Comparisons of efficiency between HOGT and other methods.
> >
> > Further analysis of the number of community nodes.
> >
> > We hope the new experimental results and explanations address your concerns.
> >
> > As the discussion deadline approaches, we kindly inquire if you have any further suggestions for improving our manuscript. Your feedback is invaluable, and we would greatly appreciate your guidance.
> >
> > If our responses have sufficiently addressed your concerns, we kindly hope you might reconsider the rating. Thank you once again for your thoughtful review and consideration.
> >
> > Best regards,
> >
> > The Authors

---

> > > ### Author Response · Authors · 2024-11-27
> > >
> > > Dear Reviewer,
> > >
> > > Thank you again for your thoughtful comments and suggestions on our paper. We hope that our responses and updates to the manuscript have adequately addressed your concerns.
> > >
> > > As the discussion period approaches its conclusion, we would greatly appreciate any additional feedback or suggestions you might have for improving the work. If there are any remaining points of concern or clarification needed, we are happy to provide further elaboration.
> > >
> > > We deeply value your insights and thank you for your time and consideration.
> > >
> > > Best regards,
> > >
> > > The Authors

---

> > > > ### Comment · Reviewer_RHuk · 2024-11-30
> > > > **Official Comment by Reviewer RHuk**
> > > >
> > > > Thank you for your rebuttal. I have no further questions. I will increase my score.

---

> ### Author Response · Authors · 2024-12-01
>
> Dear Reviewer,
>
> Thank you for your thoughtful consideration and for taking the time to review our rebuttal. We greatly appreciate your feedback and your decision to increase the score in support of our paper.
>
> Best regards,
>
> The Authors

---

### Official Review · Reviewer_nhvf · 2024-10-29

**Soundness:** 2
**Presentation:** 4
**Contribution:** 2
**Rating:** 3
**Confidence:** 3

**Summary:**

The paper presents HOGT, a graph transformer that uses community-based processing to handle graph topology and computation complexity issues.
The method consists of three parts: Community sampling using reinforcement learning, Message-passing within communities and information propagation between community nodes. HOGT achieves highly competitive results
across node and graph classification tasks.

**Strengths:**

* The technical part of the paper is good -- the method is of careful design and implementation.

**Weaknesses:**

1. The problems of node classification and graph classification are well-studied in the past 10 years.
You can find the old baselines like GAT are very competitive.  Due to task saturation, HOGT shows relatively small improvements compared to these simple algorithms.
2. The theoretical part of the paper seems like mainly from a related work. Further analysis about HOGT is needed.
3. The method is too complex.

**Questions:**

N/A

---

> ### Author Response · Authors · 2024-11-20
> **Response to Reviewer nhvf**
>
> Dear Reviewer,
>
> We sincerely appreciate your time and effort in providing valuable feedback. Below, we have addressed each of the concerns and questions raised.
>
> **Q1.**
> The problems of node classification and graph classification are well-studied in the past 10 years. You can find the old baselines like GAT are very competitive. Due to task saturation, HOGT shows relatively small improvements compared to these simple algorithms.
>
> **A1.**
> Please allow us to clarify the following points:
>
> 1. In our experiments, HOGT consistently achieves notable performance improvements across various datasets compared to traditional baselines. For example, as shown in Table 2 of the paper, HOGT outperforms GAT by absolute margins of 4.15\% and 4.46\% on Pubmed and ogbn-arxiv, respectively. Furthermore, HOGT demonstrates significantly better performance on heterophilic datasets, achieving substantial margins of improvement over GAT and other traditional GCN-based methods. Additionally, we conducted a t-test to evaluate the statistical significance of HOGT's improvements, finding that the gains over the baselines are highly significant (p-value $\ll$ 0.05).
>
> 2. Traditional GCN-based methods generally perform well on homophilic datasets (as shown in Table 2 of the paper), while heterophily-based methods like H2GCN and GPRGNN excel on heterophilic datasets (Table 3 in the paper). GT models, on the other hand, deliver superior results on large-scale datasets such as ogbn-arxiv, roman-empire, and amazon-ratings, which require capturing long-range dependencies.
>
> HOGT distinguishes itself as a unified framework capable of capturing diverse types of information—local, global, and high-order relationships (Table 1 in the paper). Unlike prior models that focus on specific aspects, HOGT demonstrates versatility by effectively accommodating various graph types (graphs and hypergraphs), data characteristics (homophily and heterophily), data scales (small-scale and large-scale), and diverse graph tasks. This adaptability underscores the broader applicability of the HOGT framework.
>
> ### Table: A summary of the capabilities of various graph models in processing graph information and types
>
> | Model          | Local Information | Global Information | Higher-Order Information | Graph | Hypergraph |
> |-----------------|-------------------|--------------------|--------------------------|-------|------------|
> | GNN            | ✓                 | ✗                  | ✗                        | ✓     | ✗          |
> | HGNN           | ✓                 | ✗                  | ✓                        | ✓     | ✓          |
> | GT             | ✓                 | ✓                  | ✗                        | ✓     | ✗          |
> | HOGT (ours)    | ✓                 | ✓                  | ✓                        | ✓     | ✓          |
>
> **Q2.**
> The theoretical part of the paper seems like mainly from a related work. Further analysis about HOGT is needed.
>
> **A2.**
> We analyze the proposed HOGT framework from two perspectives: (1) HOGT's ability to approximate self-attention as implemented in general Graph Transformers (GTs), and (2) HOGT's functionality as a high-order Graph Transformer leveraging community nodes (detailed in Appendix A.4 of the paper).
>
> **1. Approximation of Self-Attention in HOGT:**
>
> The approximation of self-attention in HOGT is demonstrated as follows:
>
> - (1) Message-Passing Neural Networks with community nodes (MPNN+CN) can act as a self-attention layer.
>
> - (2) Within our three-step message-passing framework, the combination of MPNN+CN and self-attention achieves an approximation of full self-attention in graphs.
>
> While point (1) has been established in related work [1], we focuse on demonstrating (2) in the paper. Specifically:
>
> - In the **Graph Node-to-Community Node** (G2C-MP) step, message-passing through a newly introduced community node (connected to all nodes within the community) approximates self-attention within the community, as supported by Proposition 4.1 in the paper.
>
>  - In the **Community Node-to-Community Node** (C2C-ATTN) step, a self-attention mechanism propagates information among community nodes.
>
>  - Finally, in the **Community Node-to-Graph Node** (C2G-MP) step, global information from community nodes is propagated back to graph nodes via another MPNN+CN, approximating "full" self-attention.

---

> ### Author Response · Authors · 2024-11-20
>
> Thank you very much for your patience! Here is the rest part.
>
> **2. HOGT as a Higher-Order Graph Transformer:**
> We analyze HOGT’s ability to capture higher-order representations through the role of community nodes, which function analogously to hyperedges in hypergraph convolutional networks.
>
> - **Encoding Complex Relationships:**
>   To capture higher-order correlations in complex graphs, HGCNs introduce hyperedges connecting multiple nodes. Similarly, HOGT introduces a community node for each community, representing multiple nodes sharing common properties (e.g., semantics or structural information). Like hyperedges, community nodes connect to every node within their community, enabling higher-order information encoding.
>
> - **High-Order Message-Passing:**
>
> **(Apologies for the poor readability caused by incompatible formulas. Please refer to the relevant content in Appendix A.4 in the paper for clarification.)**
>
>   Following the message-passing paradigm, HGCNs first aggregate information along hyperedges and then propagate it to nodes. In spectral-based HGCNs, convolution is defined as:
>
> $\mathbf{\Delta}=\mathbf{D}_{v}^{-1 / 2} \mathbf{S} \mathbf{W}$
>
> $\mathbf{D}_{e}^{-1} \mathbf{S}^{T}$
>
> $ \mathbf{D}_{v}^{-1 / 2}$,
>
> $\boldsymbol{h}^{(k)} =\sigma\left(\mathbf{\Delta} \boldsymbol{Z}^{(k-1)} \mathbf{\Theta}^{(k)}\right)$,
>
>  where $\mathbf{D}_v$ and $\mathbf{D}_e$ are diagonal matrices representing vertex and hyperedge degrees, $\mathbf{S}$ is the incidence matrix indicating node-hyperedge relationships, and $\mathbf{W}$ represents hyperedge connections. This can be refined into three steps: node-to-hyperedge, hyperedge-to-hyperedge, and hyperedge-to-node:
>
> $  \boldsymbol{a}_{e^h}^{(k)} = \mathbf{S}^{\top} \boldsymbol{z}^{(k-1)}, $
>
> $  \boldsymbol{a}_{e^h}^{(k)} = $
>
> $\mathbf{W} \boldsymbol{a}_{e^h}^{(k)},  $
>
> $  \boldsymbol{z}^{(k)} = \mathbf{S} \boldsymbol{a}_{e^h}^{(k)}.$
>
>   Similarly, HOGT’s three-step message-passing process—Graph Node-to-Community Node, Community Node-to-Community Node, and Community Node-to-Graph Node—mirrors this structure. Moreover, in HGCNs, the relationships of hyperedges can typically be ignored, i.e., $\mathbf{W}=\mathbf{I}$. In HOGT, the framework can also be simplified to two steps, excluding the Community Node-to-Community Node step.
>
> At a high level, graph convolutional networks can be seen as special cases of hypergraph convolutional networks. In comparison, our proposed HOGT framework can be simplified to other existing GT models, demonstrating its adaptability and generalizability.
>
> [1] Chen Cai, et al. On the connection between MPNN and graph transformer. ICML 2023.
>
> **Q3.**
> The method is too complex.
>
> **A3.**
> While HOGT employs a general framework with a multi-step message-passing strategy, its overall complexity remains low and manageable.
>
> **1. Computational Complexity:**
> The complexity of HOGT is $\mathcal{O}(m^2 + N)$, where $m$ is the number of communities and $N$ is the number of graph nodes. Since $m \ll N$, the complexity of HOGT approximates to $\mathcal{O}(N)$, making it nearly linear. In contrast, general Graph Transformers (GTs) have a complexity of $\mathcal{O}(N^2)$. This difference makes HOGT significantly more computationally efficient than general GTs, particularly for large-scale graphs.
> Regarding the time complexity of community sampling, this process is performed as a preprocessing step using techniques like random walk or spectral clustering and does not incur additional computational costs during model training. Furthermore, HOGT maintains excellent performance even when treating the entire graph as a single community, demonstrating its flexibility.
>
> **2. Efficiency in Practice:**
> Experimental results confirm that HOGT significantly reduces training time and memory usage, addressing concerns about its complexity. Table 6 in the paper reports the training time per epoch, inference time, and GPU memory consumption on the Cora and ogbn-arxiv datasets. To ensure a fair comparison, we report the training time per epoch, as fixed training epochs are standard for these datasets.
> We observe that HOGT is orders of magnitude faster than popular GT models, including Graphormer, LiteGT, and Polynormer. Additionally, HOGT's memory consumption is substantially lower due to its simplified global attention mechanism, which scales with $\mathcal{O}(N)$ complexity.
>
> These aspects demonstrate that while HOGT introduces a general and powerful framework, its design is computationally efficient, practical, and well-suited for large and complex graph datasets.

---

> > ### Author Response · Authors · 2024-11-25
> >
> > Dear Reviewer,
> >
> > Thanks a lot to you for your time and effort in reviewing our paper. For each question you have raised, we have thoughtfully provided our explanations and hope the explanations can alleviate your uncertainty.
> >
> > As the discussion deadline is approaching, we would like to inquire if you have any further suggestions for improving our manuscript. We would greatly value your input and appreciate your guidance.
> >
> > If our responses have sufficiently addressed your concerns, we kindly hope you might reconsider the rating. Thank you for your time and consideration.
> >
> > Best Regards,
> >
> > Authors

---

> > > ### Author Response · Authors · 2024-12-02
> > >
> > > We deeply appreciate the valuable feedback from all reviewers. We are pleased to note that during the rebuttal phase, we successfully addressed the concerns of other reviewers, which led to either maintained positive evaluations or improved scores. We genuinely hope our detailed and thoughtful response will also resolve your concerns and help improve the score of our paper. We sincerely look forward to your feedback and thank you once again for your time and consideration!

---

> > > > ### Author Response · Authors · 2024-12-03
> > > > **Response to Reviewer nhvf**
> > > >
> > > > Dear Reviewer nhvf,
> > > >
> > > > Thank you once again for your review. We are pleased to see your recognition of the strengths in the technical aspects of our work.
> > > >
> > > > In the updated manuscript, we have further demonstrated the effectiveness of the proposed HOGT across a variety of tasks, including the addition of a link prediction task (Appendix A.9) and an evaluation of its robustness on sparse graph structures (Appendix A.11). During the rebuttal phase, we successfully addressed the concerns of other reviewers, which led to either maintained positive evaluations or improved scores. We genuinely hope our detailed and comprehensive response will also address your concerns and contribute to an improved evaluation of our paper.
> > > >
> > > > Thank you once again for your time and consideration. We sincerely look forward to your feedback.
> > > >
> > > > Best regards,
> > > >
> > > > The Authors

---

### Official Review · Reviewer_X76c · 2024-11-01

**Soundness:** 3
**Presentation:** 3
**Contribution:** 3
**Rating:** 6
**Confidence:** 4

**Summary:**

The paper proposes a high-order graph transformer (HOGT) for graph learning tasks. HOGT introduces a flexible sampling method to extract communities from the graph and a three-step message-passing strategy to capture local, long-range, and higher-order relationships of the graph. The paper demonstrates the effectiveness of HOGT on node classification tasks and shows its superiority over other graph models.

**Strengths:**

(1) The paper introduces a novel approach, HOGT, that combines community-based sampling and message-passing to capture comprehensive information in graph learning.

(2) HOGT achieves competitive results on various graph datasets, demonstrating its effectiveness in node classification tasks.

(3) The paper provides a theoretical analysis of HOGT, showing its approximation capabilities and the relationship with other graph models.

**Weaknesses:**

(1) Domain Limitation of Datasets. Expanding the evaluation to include diverse domains, such as those in the TEG-DB datasets [1], which feature rich node and edge text, would strengthen the findings.

(2) Narrow Applicability. The model’s applicability is somewhat restricted to specific tasks within graph domains, such as node classification. The authors should consider its potential for other important tasks, like link prediction.

[1] "TEG-DB: A Comprehensive Dataset and Benchmark of Textual-Edge Graphs." NeurIPS 2024.

**Questions:**

See weakness above.

---

> ### Author Response · Authors · 2024-11-20
> **Response to Reviewer X76c**
>
> Dear Reviewer,
>
> Thank you for taking the time to review our paper and provide valuable feedback. Below are our responses to your concerns.
>
> **Q1.**
> Domain Limitation of Datasets. Expanding the evaluation to include diverse domains, such as those in the TEG-DB datasets [1], which feature rich node and edge text, would strengthen the findings.
>
> **A1.**
> Thank you for your insightful suggestion. We have conducted experiments on the TEG-DB datasets [1], specifically the Goodreads-Children and Goodreads-Crime datasets. The results, summarized in the table below, demonstrate that HOGT achieves either better or comparable performance compared to other methods. In these experiments, we treated the data in each batch as a community and introduced a community node for each batch, effectively extending the HOGT framework to these diverse domains.
>
> **Table: Performance Comparison Across Methods on Goodreads-Children.**
>
> | Methods           | AUC (BERT-Large)  | F1 (BERT-Large) | AUC (BERT-Base) | F1 (BERT-Base) | AUC (w/o Edge Text) | F1 (w/o Edge Text) |
> |--------------------|----------------|---------------|---------------|--------------|-------------------|------------------|
> | GeneralConv       | 0.9810         | 0.9179        | 0.9821        | 0.9187       | 0.9825            | 0.9189           |
> | GraphTransformer  | 0.9807         | 0.9200        | 0.9811        | 0.9160       | 0.9776            | 0.9066           |
> | HOGT              | **0.9821**     | **0.9216**    | **0.9837**    | **0.9208**   | **0.9825**        | **0.9289**       |
>
> **Table: Performance Comparison Across Methods on Goodreads-Crime.**
>
> | Methods           | AUC (BERT-Large) | F1 (BERT-Large) | AUC (BERT-Base) | F1 (BERT-Base) | AUC (w/o Edge Text) | F1 (w/o Edge Text) |
> |--------------------|----------------|---------------|---------------|--------------|-------------------|------------------|
> | GeneralConv       | 0.9772         | 0.9079        | 0.9774        | 0.9077       | 0.9752            | 0.9101           |
> | GraphTransformer  | 0.9738         | 0.9079        | 0.9737        | 0.9079       | 0.9716            | 0.8983           |
> | HOGT              | **0.9776**     | **0.9110**    | **0.9776**    | **0.9110**   | **0.9768**        | **0.9130**       |
>
> [1] Zhuofeng Li, et al. TEG-DB: A Comprehensive Dataset and Benchmark of Textual-Edge Graphs. NeurIPS 2024.
>
> **Q2.**
> Narrow Applicability. The model’s applicability is somewhat restricted to specific tasks within graph domains, such as node classification. The authors should consider its potential for other important tasks, like link prediction.
>
> **A2.**
> In addition to the experiments on the TEG-DB datasets for link prediction, we also applied HOGT to graph classification tasks to further demonstrate its versatility and superiority, as detailed in Appendix A.8.
>
> **Performance on Graph Classification**
>
> We evaluated HOGT on several widely-used real-world datasets from the TU database [2] for graph classification tasks.
>
> NCI1: This dataset consists of 4,110 molecular graphs representing two balanced subsets of chemical compounds screened for activity against non-small cell lung cancer and ovarian cancer cell lines.
>
> PROTEINS: This dataset includes 1,113 protein graphs, where each graph corresponds to a protein molecule. Nodes represent amino acids, and edges capture interactions between them.
>
> As shown in Table 12 in the Appendix A.8, HOGT achieves state-of-the-art performance across all datasets. Compared to GT models such as GraphGPS, HOGT demonstrates the ability to encode more comprehensive and nuanced information in the graph, highlighting its effectiveness and broader applicability beyond node classification.
>
> [2] Christopher Morris, et al. Tudataset: A collection of benchmark datasets for learning with graphs. arXiv:2007.08663, 2020.

---

> > ### Author Response · Authors · 2024-11-25
> >
> > Dear Reviewer,
> >
> > Thank you very much for your time and effort in reviewing our paper. Based on your feedback, we have added more experiments on other domains (TEG-DB) for the link prediction task to further demonstrate the effectiveness of HOGT in the updated manuscript (highlighted in blue).
> >
> > As the discussion deadline approaches, we kindly inquire if you have any further suggestions for improving our manuscript. We deeply value your input and would greatly appreciate your guidance.
> >
> > Thank you once again for your time and consideration.
> >
> > Best regards,
> >
> > The Authors

---

> > > ### Comment · Reviewer_X76c · 2024-11-26
> > >
> > > Thanks for your response!

---

> > > > ### Author Response · Authors · 2024-12-03
> > > >
> > > > Dear Reviewer,
> > > >
> > > > Thank you for your thoughtful review and for taking the time to consider our rebuttal. We are pleased to see that we have adequately addressed your concerns, with no further questions remaining.
> > > >
> > > > As the discussion deadline approaches, and given that all comments have been thoroughly addressed, we kindly request you to reconsider your rating. Once again, thank you for your insightful review and valuable time.
> > > >
> > > > Best regards,
> > > >
> > > > The Authors

---

### Official Review · Reviewer_2D8Y · 2024-11-03

**Soundness:** 3
**Presentation:** 3
**Contribution:** 3
**Rating:** 6
**Confidence:** 2

**Summary:**

This paper introduces HOGT (High-Order Graph Transformer), a new architecture that tackles key issues in existing graph transformers, especially around capturing topology and scaling to large graphs. The authors use a three-step message-passing process: sampling communities from the graph, creating community nodes as information bridges, and enabling message flow between graph and community nodes. This design removes the need for positional encoding, embedding structure naturally through communities. HOGT shows strong performance across different types of graphs, with impressive computational efficiency.

**Strengths:**

1. The paper introduces a well-founded architecture with a three-step message-passing strategy that effectively captures multi-scale information in graphs, handling local, global, and higher-order details. Theoretically, it’s shown that HOGT can approximate global attention and unify existing models, while the community-based design removes the need for positional encoding.

2. HOGT also demonstrates strong versatility, performing well on various graph types (homophilic, heterophilic, and hypergraphs) and adapting to different community sampling methods, which enhances scalability across graph sizes. Efficiency is greatly improved, with computational complexity reduced from O(N²) to O(m² + N), validated by experiments that show strong results over state-of-the-art methods, especially on challenging datasets.

**Weaknesses:**

1. The strict hierarchy in the three-step message-passing mechanism could introduce bottlenecks in information flow. By requiring all long-range communication to route through community nodes, the model risks distorting or weakening critical direct relationships between nodes—especially in tasks where pairwise connections hold essential information. The assumption that this hierarchical structure is universally beneficial may be too broad, as the paper offers little discussion on cases where direct node-to-node communication might better capture necessary details.

2. The approach to initializing community nodes also feels underdeveloped and could pose challenges. Starting with random initialization may lead to instability and slower convergence, particularly in early training stages. Additionally, there's no clear strategy for aligning community node dimensionality with original node features, which seems like a significant gap. Given that these community nodes are crucial bridges for information flow, their initial setup could substantially impact the quality of the representations learned.

**Questions:**

See weakness

---

> ### Author Response · Authors · 2024-11-20
> **Response to Reviewer 2D8Y**
>
> Dear Reviewer:
>
> Thank you for your constructive comments and suggestions. They have been invaluable in helping us enhance the quality and clarity of our paper. Please find our point-by-point responses to your concerns below.
>
> **Q1.**
> The strict hierarchy in the three-step message-passing mechanism could introduce bottlenecks in information flow. By requiring all long-range communication to route through community nodes, the model risks distorting or weakening critical direct relationships between nodes—especially in tasks where pairwise connections hold essential information. The assumption that this hierarchical structure is universally beneficial may be too broad, as the paper offers little discussion on cases where direct node-to-node communication might better capture necessary details.
>
> **A1.**
> Please allow us to clarify pairwise connections and their importance from the following perspectives:
>
> **Preservation of Pairwise Connections:**
> In scenarios where pairwise relationships hold critical information, the proposed HOGT framework explicitly addresses this during the final step—Community Node-to-Graph Node. At this stage, the representation of each graph node is updated by aggregating information from both its associated community nodes and its directly connected neighbors. This mechanism ensures that essential local connections are preserved and effectively incorporated into the model.
>
> **Flexibility in Community Optimization:**
> HOGT is a general framework that allows for the optimization of the number of communities to suit different datasets. In extreme cases, treating the entire dataset as a single community naturally emphasizes direct node-to-node communication. Furthermore, the random walk sampling approach generates communities for only a subset of graph nodes, offering flexibility in balancing local and global interactions. This adaptability ensures that the model can capture pairwise relationships when necessary while still benefiting from the hierarchical structure.
>
> Overall, the hierarchical structure introduced by community nodes provides a versatile framework for balancing local and global interactions in graphs. While HOGT is designed to capture comprehensive information across various types of graphs, we acknowledge that achieving an optimal balance between local and global information in complex graph structures remains a challenge. This is a current limitation and a promising direction for future research.
>
> **Q2.**
> The approach to initializing community nodes also feels underdeveloped and could pose challenges. Starting with random initialization may lead to instability and slower convergence, particularly in early training stages. Additionally, there's no clear strategy for aligning community node dimensionality with original node features, which seems like a significant gap. Given that these community nodes are crucial bridges for information flow, their initial setup could substantially impact the quality of the representations learned.
>
> **A2.**
> We appreciate the reviewer’s insightful comments regarding the initialization of community nodes and its potential impact on stability and convergence.
>
> **Initialization of Community Nodes:**
> The introduced virtual nodes (community nodes) can be initialized using either zero vectors or random initialization. In our experiments, we observed that both approaches resulted in similar final performance after 200 epochs, suggesting that the choice of initialization method has minimal impact on the final outcomes.
>
> **Stability and Convergence:**
> To address your concerns about stability, we have conducted experiments with 10 independent runs using different random seeds. The results demonstrated low variance, indicating that our framework is robust and stable across various initialization conditions.
> While random or zero initialization is effective, an alternative approach—such as using max or mean pooling of the features of graph nodes within a community to initialize community node features—could potentially accelerate convergence by providing a more informed starting point for community node embeddings. However, this method introduces additional computational overhead, which we have deliberately avoided in our current implementation to maintain efficiency.
>
> We will update our paper to discuss the potential advantages of alternative initialization strategies and the trade-offs involved, ensuring a comprehensive understanding of our approach.

---

> > ### Author Response · Authors · 2024-11-25
> >
> > Dear Reviewer,
> >
> > Thank you very much for your time and effort in reviewing our paper. Based on your feedback, we have made several updates to the manuscript (highlighted in blue), including:
> >
> > Adding a limitation analysis of HOGT in the Conclusion.
> >
> > Including an analysis of community node initialization.
> >
> > As the discussion deadline approaches, we kindly inquire if you have any further suggestions for improving our manuscript. We deeply value your input and would greatly appreciate your guidance.
> >
> > Thank you once again for your time and consideration.
> >
> > Best regards,
> >
> > The Authors

---

> ### Comment · Reviewer_2D8Y · 2024-11-25
>
> Thank you for your rebuttal. I have no further questions. I will maintain my score.

---

> > ### Author Response · Authors · 2024-12-03
> >
> > Dear Reviewer,
> >
> > Thanks once again for supporting our paper with a positive score. While you have no questions about this paper, we noticed that your confidence level remains low. As the discussion period nears its conclusion, we kindly hope you to consider reevaluating.
> >
> > Best Regards,
> >
> > The Authors

---

### Author Response · Authors · 2024-12-03

Dear Chairs and Reviewers,

Hope this message finds you well.

With the closing of the discussion period, we present a brief summary of our discussion with the reviewers as an overview for reference. First of all, we thank all the reviewers for their insightful comments and suggestions. We are encouraged by the positive feedback, as highlighted below:

R1. "The paper introduces a well-founded architecture...", "Theoretically, it’s shown that HOGT can approximate global attention and unify existing models,...", "demonstrates strong versatility, efficiency, ...".

R2. "The paper introduces a novel approach...", "demonstrating its effectiveness in node classification tasks", "provides a theoretical analysis of HOGT,...".

R3. "The technical part of the paper is good -- the method is of careful design and implementation."

R4. “The idea is novel”， “contributes to the advancement of the field and opens up new research directions for further exploration.“.

We have carefully addressed all the comments and provided detailed responses. Since we did not receive specific questions or response from Reviewer nhvf during rebuttal, we summarize the main concerns of other reviews and outline the corresponding updates in the revised manuscript:

**The approach to initializing community nodes.** We added an analysis of community node initialization in Appendix A.7 and shown that the existing initialization approach is reasonable.

**The performance of HOGT on link prediction.** Additional experiments on the TEG-DB dataset for link prediction are included in Appendix A.9, further highlighting the effectiveness of HOGT.

**Hyperparameter sensitivity and robustness evaluations.** We conducted hyperparameter sensitivity and robustness evaluations, presented in Appendices A.10 and A.11, respectively. The results confirm the robustness of the proposed HOGT.

Based on the discussion with reviewers, we also present a brief summary of our paper as follows.

**Observation:** Existing graph models struggle to effectively capture the complex structural relationships in the graph for different graphs and data types, while also providing theoretical support.

**Solution:** We propose a flexible sampling method followed by a three-step message-passing framework in GTs to capture comprehensive information achieving high expressiveness for graph representation learning.

**Results:** The effectiveness of our framework has been demonstrated on various graph types (graph and hypergraph), data types (homophily and heterophily), data scales (same-scale and large-scale), and different graph tasks (node classification, graph classification, and link prediction).

**Highlights:**

- Introduced a higher-order message-passing strategy with flexible sampling methods.

- Unified message-passing and GTs by constructing communities and introducing new community nodes.

- Provided theoretical proof that the three-step message-passing framework with newly introduced community nodes achieves global attention akin to general transformers.

- Demonstrated the versatility and robustness of HOGT through extensive experiments.


Thanks again for your efforts in the reviewing and discussion. We appreciate all the valuable feedback that helped us to improve our submission.

Sincerely

The Authors

---

### Meta-Review · Area_Chair_o8UW · 2024-12-20

**Metareview:**

- Scientific Claims and Findings:
    - This paper introduces a high-order graph transformer (HOGT) architecture designed for node classification. The approach involves sampling communities from the graph, creating community nodes, and facilitating message passing between the graph and community nodes. HOGT demonstrates competitive performance across various node classification tasks.
- Strengths:
   - The introduction of a learnable community sampling method using reinforcement learning.
   - The design of the proposed HOGT is well-reasoned.
- Weaknesses:
    - Although HOGT is a reasonable design, it closely resembles existing works such as hierarchical graph transformers or those utilizing graph cluster structures [1] [2] [3] [4], as mentioned in lines 144–151 of the related work section. While the authors discuss the distinctions between HOGT and these works, the core differences do not appear to be substantial.

             [1] Wenhao Zhu, Tianyu Wen, Guojie Song, Xiaojun Ma, and Liang Wang. Hierarchical transformer for scalable graph learning, 2023.

             [2] Wenhao Zhu, Guojie Song, Liang Wang, and Shaoguo Liu. Anchorgt: Efficient and flexible attention architecture for scalable graph transformers, 2024.

             [3] Weirui Kuang, Z WANG, Yaliang Li, Zhewei Wei,and Bolin Ding. Coarformer: Transformer for large graph via graph coarsening, 2022.

             [4] Yujie Xing, Xiao Wang, Yibo Li, Hai Huang, and Chuan Shi. Less is more: on the over-globalizing problem in graph transformers, 2024.

   - The AC concurs with Reviewer nhvf that similar performance can be achieved by older and simpler models. Consequently, the advantages offered by HOGT and similar graph transformer methods may not be significant.

- Most Important Reasons for Decision:
     - Based on the weaknesses mentioned above.

**Additional Comments On Reviewer Discussion:**

In their rebuttal, the authors presented additional experimental results on graph tasks such as link prediction and graph classification, along with a more detailed sensitivity analysis of hyperparameters.

After the rebuttal, Reviewers 2D8Y and X76c maintained their ratings at 6, Reviewer Rhuk increased their rating from 5 to 6, and Reviewer nhvf kept their rating at 3.

---

### Decision · Program_Chairs · 2025-01-22

Reject